# NuRD-interacting protein ZFP296 regulates genome-wide NuRD localization and differentiation of mouse embryonic stem cells

Susan L. Kloet [1,3], Ino D. Karemaker [2], Lisa van Voorthuijsen [2], Rik G.H. Lindeboom [2], Marijke P. Baltissen[2], Raghu R. Edupuganti[1], Deepani W. Poramba-Liyanage [1], Pascal W.T.C. Jansen[2] & Michiel Vermeulen [2]

The nucleosome remodeling and deacetylase (NuRD) complex plays an important role in gene expression regulation, stem cell self-renewal, and lineage commitment. However, little is known about the dynamics of NuRD during cellular differentiation. Here, we study these dynamics using genome-wide profiling and quantitative interaction proteomics in mouse embryonic stem cells (ESCs) and neural progenitor cells (NPCs). We find that the genomic targets of NuRD are highly dynamic during differentiation, with most binding occurring at cell-type specific promoters and enhancers. We identify ZFP296 as an ESC-specific NuRD interactor that also interacts with the SIN3A complex. ChIP-sequencing in *Zfp296* knockout (KO) ESCs reveals decreased NuRD binding both genome-wide and at ZFP296 binding sites, although this has little effect on the transcriptome. Nevertheless, *Zfp296* KO ESCs exhibit delayed induction of lineage-specific markers upon differentiation to embryoid bodies. In summary, we identify an ESC-specific NuRD-interacting protein which regulates genome-wide NuRD binding and cellular differentiation.

[1] Department of Molecular Biology, Faculty of Science, Radboud Institute for Molecular Life Sciences, Radboud University Nijmegen, Nijmegen, 6500 HB The Netherlands. [2] Department of Molecular Biology, Faculty of Science, Radboud Institute for Molecular Life Sciences, Oncode Institute, Radboud University Nijmegen, Nijmegen, 6500 HB The Netherlands. [3] Present address: Leiden Genome Technology Center, Department of Human Genetics, Leiden University Medical Center, Leiden, 2300 RC The Netherlands. These authors contributed equally: Susan L. Kloet, Ino D. Karemaker.  Correspondence and requests for materials should be addressed to M.V. (email: michiel.vermeulen@science.ru.nl)

The nucleosome remodeling and deacetylase (NuRD) complex is an evolutionarily conserved chromatin-associated protein complex which regulates gene expression and also plays a role in the DNA damage response[1–3]. The complex contains two enzymatic functions: histone deacetylase activity, catalyzed by HDAC1 and HDAC2, and ATP-dependent chromatin remodeling activity, catalyzed by CHD3, CHD4, or CHD5. Other core subunits of the complex include DOC-1 (also known as CDK2AP1), GATAD2A and -B, RBBP4 and -7, MTA1, -2, and -3, and MBD2 and -3. Some of these paralogous subunits define mutually exclusive NuRD subcomplexes with distinct biological functions[4,5]. In addition, NuRD has been shown to interact with a large number of proteins such as FOG1, SALL4, JUN, and Ikaros, some of which serve to recruit NuRD to specific target sites in the genome[6–9].

Due to the presence of the HDAC1/2 subunits, NuRD can be categorised as part of the HDAC1/2 complex family, other members of which are the SIN3 and CoREST complexes[10]. Although HDAC1/2 complexes have traditionally been classified as transcriptional co-repressor complexes, recent genome-wide analyses revealed that NuRD is mainly associated with promoters and enhancers of genes that are actively being transcribed. The exact role of NuRD in regulating gene expression is still not completely understood, but one hypothesis is that NuRD mainly serves to fine-tune expression levels of target genes rather than enabling stable gene repression[11–13].

Apart from its functions in gene expression and the DNA damage response, the NuRD complex also regulates cell fate and lineage commitment during early development, and has been reported to be part of the embryonic stem cell pluripotency network[14–16]. As such, numerous studies have investigated the composition and genome-wide profile of the NuRD complex in embryonic stem cells (ESCs). Yet less is known about the dynamics of the NuRD complex, both at the genomic and proteomic level, during ESC differentiation.

Here, we perform an integrative proteomic and genomic characterization of the MBD3/NuRD complex in undifferentiated mouse ESCs as well as neural progenitor cells (NPCs), which we obtain through in vitro differentiation of ESCs[17]. Our data reveal that the genome-wide binding of MBD3/NuRD is highly dynamic during differentiation, with most ESC-specific binding occurring at promoters and enhancers. MBD3/NuRD affinity purifications followed by mass spectrometry in ESCs and NPCs identify zinc finger protein 296 (ZFP296) as a prominent, stem cell-specific NuRD interactor. Reciprocal ZFP296 purifications confirm this interaction and reveal that ZFP296 is a shared interactor of the NuRD and SIN3A complexes in ESCs. Knockout (KO) of *Zfp296* in ESCs leads to a decrease in NuRD binding, both genome-wide as well as at ZFP296 target genes. Additionally, the expression of several lineage commitment genes is perturbed in the absence of ZFP296 in ESCs, and we observe that *Zfp296* KO ESCs display delayed differentiation upon withdrawal of leukaemia inhibitory factor (LIF). In summary, we identify ZFP296 as a stem cell-specific NuRD-interacting protein which regulates genome-wide NuRD localization and differentiation of ESCs.

## Results

**NuRD binding is highly dynamic during differentiation**. To investigate the genome-wide DNA binding dynamics of the NuRD complex during mouse ESC differentiation, we performed chromatin immunoprecipitation followed by deep sequencing (ChIP-seq) using antibodies against two endogenous NuRD subunits, MBD3 and CHD4, in both ESCs and NPCs. In ESCs, 1585 binding sites for MBD3 were identified in two biological replicates; the large majority of MBD3 sites (1354) also co-localized with CHD4 peaks, indicating that these are genuine NuRD binding sites (Fig. 1a). ChIP-seq analysis of CHD4 identified a large number of peaks (7262) that did not overlap with MBD3, which is in agreement with recent data and suggests that these could be sites where CHD4 acts independently of NuRD[13,18]. A similar distribution of MBD3 and CHD4 sites was obtained in NPCs (Fig. 1a). Comparing binding sites in ESCs and NPCs revealed a surprisingly limited overlap, suggesting that many NuRD binding sites (>95%) are cell-type specific (Fig. 1a–c; Supplementary Fig. 1a). NuRD binding sites that are shared between the two cell types are enriched for transcription start sites (TSS) and are marked with H3K4me3 and H3K27ac in both ESCs and NPCs, suggesting that these occur at the promoters of constitutively active genes (Fig. 1d, e). Interestingly, the H3K27ac signal here shows a single peak rather than a double peak, which is in agreement with recent data on H3K27ac levels at NuRD-bound loci, where a single or double H3K27ac peak is characteristic of promoters or active enhancers, respectively[13]. In contrast, cell-type specific NuRD binding sites mostly (72% for NPC and 82% for ESC) map to intergenic and intronic regions (Fig. 1d). ESC-specific NuRD sites are marked with H3K27ac, H3K4me1, H3K4me3, and p300[19], indicating that dynamic NuRD binding likely occurs at active promoters and enhancers[19,20] (Fig. 1e). Remarkably, NPC-specific NuRD sites are not marked with any of the histone marks that we examined, but do seem to be highly methylated in both cell types (Fig. 1e; Supplementary Fig. 1b). The SOX2 and POU5F1/OCT4 DNA binding motifs are enriched under ESC-specific NuRD binding sites (Supplementary Fig. 1c) and genes nearby ESC-enriched peaks are involved in regulating development (Supplementary Fig. 1d, Supplementary Data 1). These findings support previous studies that revealed a role for NuRD in regulation of the pluripotency network[14–16]. NPC-specific NuRD binding sites are strongly enriched for the FOS/JUN DNA binding motif (Supplementary Fig. 1c). This observation is in agreement with a recent study, which showed that the transactivation domain of c-JUN can recruit the NuRD complex to AP-1 target genes[8]. Consistent with our findings that shared NuRD binding sites occur at the TSSs of constitutively active genes, we found that genes nearby shared NuRD peaks were enriched for GO terms related to housekeeping functions (Supplementary Fig. 1d). In summary, these experiments revealed that NuRD binding is highly dynamic during cellular differentiation and that these dynamic NuRD sites map to promoters and putative enhancers in ESCs.

**ZFP296 is an ESC-specific NuRD interactor**. Next, we set out to identify cell-type specific NuRD subunits or interactors, which could potentially affect NuRD binding to target genes in a cell-type specific manner, thereby explaining the observed dynamic binding. To this end, we created an MBD3-GFP expressing ESC line using bacterial artificial chromosome (BAC) TransGeneOmics[21], and also differentiated this line in vitro to NPCs. Importantly, expression levels of the MBD3-GFP transgene in these cell lines are similar to or lower than the expression level of the endogenous MBD3 protein (Supplementary Fig. 2a, Supplementary Fig. 4). Nuclear extracts from MBD3-GFP expressing ESCs and NPCs were subjected to GFP affinity enrichment in triplicate using GFP nanobodies. Affinity enriched proteins were then on-bead digested and analysed by nLC-MS/MS. As shown in Fig. 2a, b and Supplementary Fig. 2b, we identified a large number of overlapping MBD3-interacting proteins in ESCs and NPCs. We then calculated the intensity-based absolute quantification (iBAQ) values of the most predominant and statistically significant MBD3-interacting proteins in both

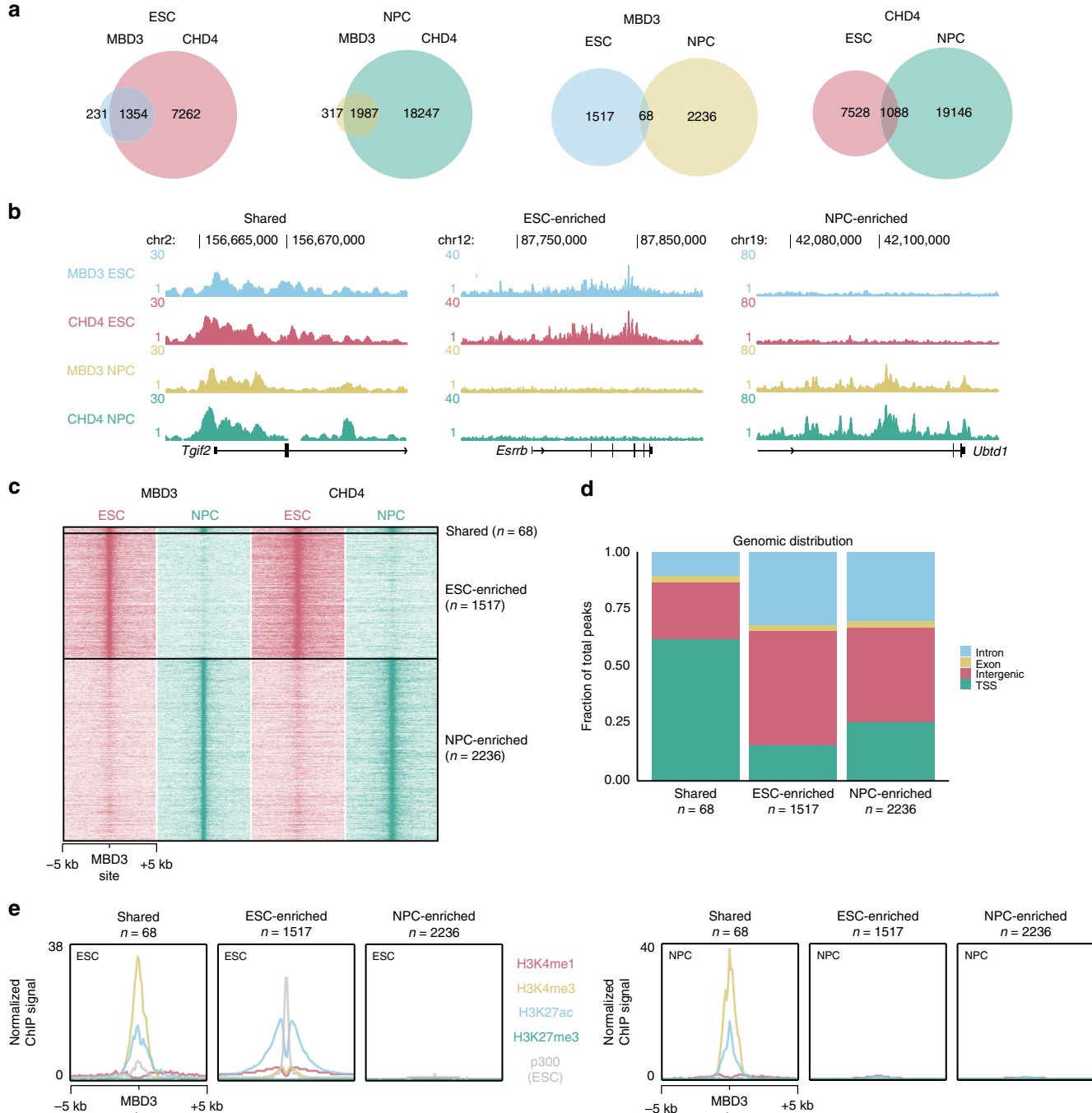

**Fig. 1** Dynamic NuRD binding sites map to active promoters and enhancers in ESCs. **a** Venn diagrams summarizing NuRD ChIP-seq results in embryonic stem cells (ESC) and neural progenitor cells (NPC). **b** UCSC genome browser screenshots of example loci where NuRD binding is shared between ESC and NPC (left), enriched in ESC (middle), or enriched in NPC (right). **c** Heat map showing the ChIP-seq read density for MBD3 and CHD4 in ESC and NPC, centred on the union of MBD3 peaks from ESC and NPC. **d** Genomic distribution of NuRD ChIP-seq peaks from each class as shown in **b** and **c**. TSS peaks were +5 kb/−1 kb from a transcription start site. **e** Band plots of four different histone marks and p300 (in ESCs, publicly available data[19]) ChIP-seq at MBD3 binding sites for each class of NuRD binding. See also Supplementary Fig. 1

cell types, which can be used to estimate the relative abundance (stoichiometry) in affinity purifications[22] (Supplementary Data 2). This analysis revealed several ESC-enriched MBD3 interactors (Fig. 2c; Supplementary Fig. 2c) including the SALL family of proteins, which have been described previously as stem cell-specific NuRD interactors[23,24]. The SALL proteins are in fact core components of the NuRD complex in ESCs (Fig. 2c; Supplementary Fig. 2c, d, Supplementary Fig. 4) as their relative abundance is nearly 1:1 with MBD3. The most ESC-enriched

MBD3 interactor is ZFP296 (Fig. 2c; Supplementary Fig. 2b, c), a relatively uncharacterized protein that carries six putative DNA binding zinc fingers and has been proposed to act as a transcription factor[25]. Interestingly, ZFP296 is a known marker protein for pluripotency and has been shown to stimulate iPSC reprogramming driven by OCT4, SOX2, KLF4 and c-MYC (OSKM)[25]. Purification of MBD3-GFP from the nuclear pellet fraction obtained after nuclear extraction revealed that the SALL4 and ZFP296 interaction with NuRD was reduced in tightly

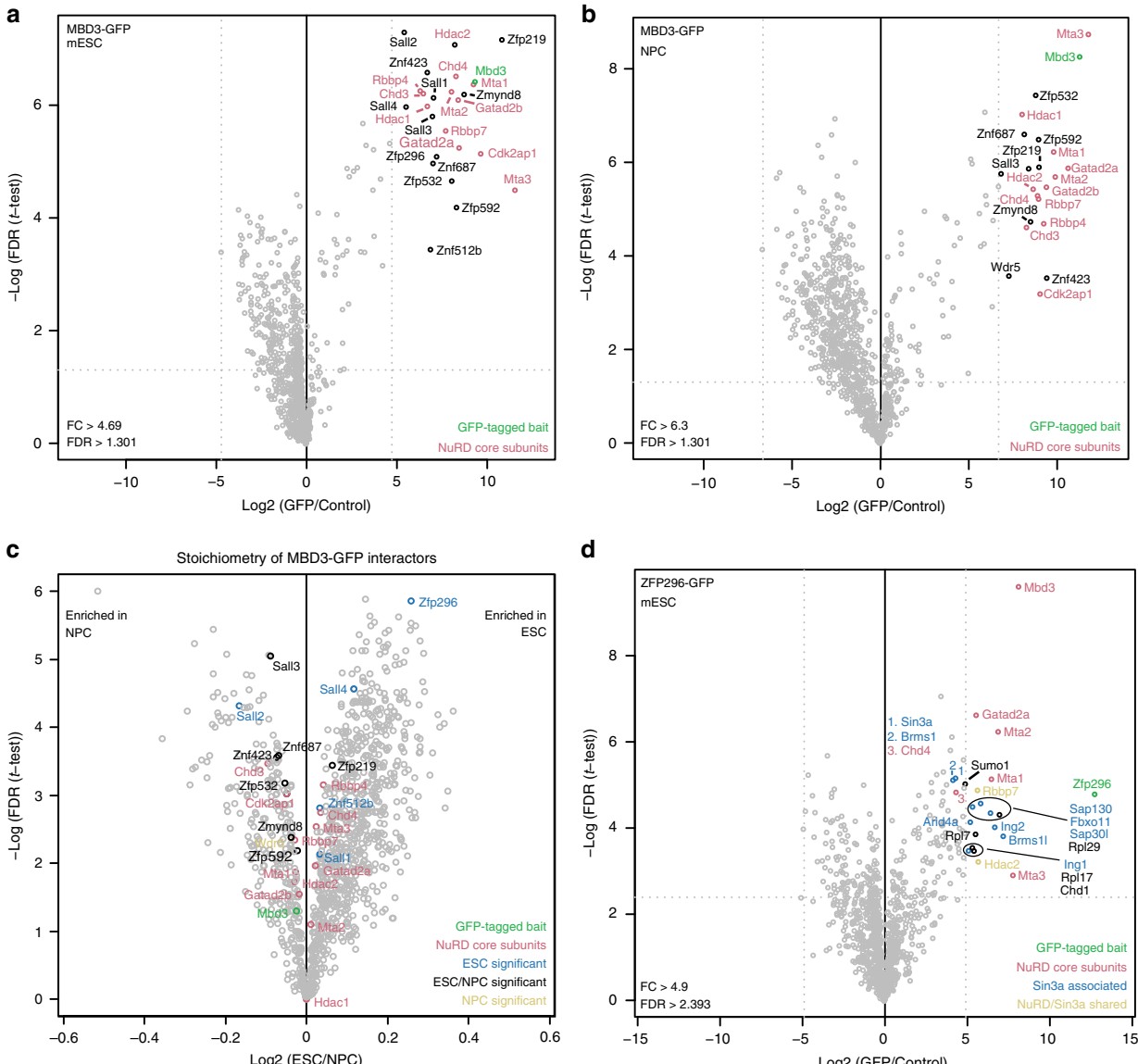

**Fig. 2** ZFP296 is an ESC-specific NuRD interactor. **a**, **b** Volcano plots from label-free GFP pulldowns on MBD3-GFP ESC (**a**) and MBD3-GFP NPC (**b**) nuclear extracts. Statistically enriched proteins in the MBD3-GFP pulldowns are identified by a permutation-based FDR-corrected $t$-test. The label-free quantification (LFQ) intensity of the GFP pulldown relative to the control [fold change (FC), $x$-axis] is plotted against the $-\log_{10}$-transformed $P$-value of the $t$-test ($y$-axis). The proteins in the upper right corner represent the bait (MBD3, green) and its interactors. NuRD core subunits are colour-coded in red. **c** Volcano plot of the stoichiometry of MBD3-GFP interactors in ESCs and NPCs. The iBAQ value of each protein group is divided by the iBAQ value of the core NuRD subunit HDAC1/2, then graphed as in **a**. Proteins that were identified as significant outliers in **a** or **b** are colour-coded in blue or yellow, respectively, and those that were significant in both are in black. **d** Volcano plot of GFP-ZFP296 interacting proteins in ESCs graphed as in **a**. NuRD core subunits are colour-coded in red, SIN3A associated proteins are in blue, and proteins that are shared subunits between NuRD and SIN3A are in yellow. See also Supplementary Fig. 2

chromatin-bound NuRD compared to lightly chromatin-bound or unbound NuRD (Supplementary Fig. 2e, f). The iBAQ value of ZFP296 relative to HDAC1/2 in the MBD3-GFP pulldown is ~ 0.4 (Supplementary Fig. 2c), suggesting that ZFP296 is a prominent NuRD interactor in the soluble nuclear fraction. To verify the detected interaction between NuRD and ZFP296, we generated an ESC line expressing ZFP296 with a GFP tag. Nuclear extracts from this cell line were subjected to GFP affinity purifications followed by nLC-MS/MS (Fig. 2d) or immunoblotting (Supplementary Fig. 2g, Supplementary Fig. 4). These experiments confirmed that ZFP296 interacts with the NuRD complex in ESCs. Additionally, subunits of the SIN3A complex were identified as statistically significant ZFP296 interactors, indicating

that ZFP296 interacts with both the NuRD and SIN3A complexes in ESCs (Fig. 2d; Supplementary Fig. 2h). Taken together, these experiments revealed that certain NuRD-interacting proteins display cell-type specific interaction dynamics.

**ZFP296 co-localizes with NuRD and SIN3A in ESCs.** To investigate the putative function of ZFP296 as a NuRD- and SIN3A-interacting protein, we performed ChIP-seq on the GFP-tagged ZFP296 ESC line using a GFP antibody. 3102 GFP-ZFP296 peaks were identified, and many of these overlap with NuRD subunits MBD3 and CHD4, and SIN3A subunit SIN3A (21%, 42%, and 18% of significantly called peaks, respectively)

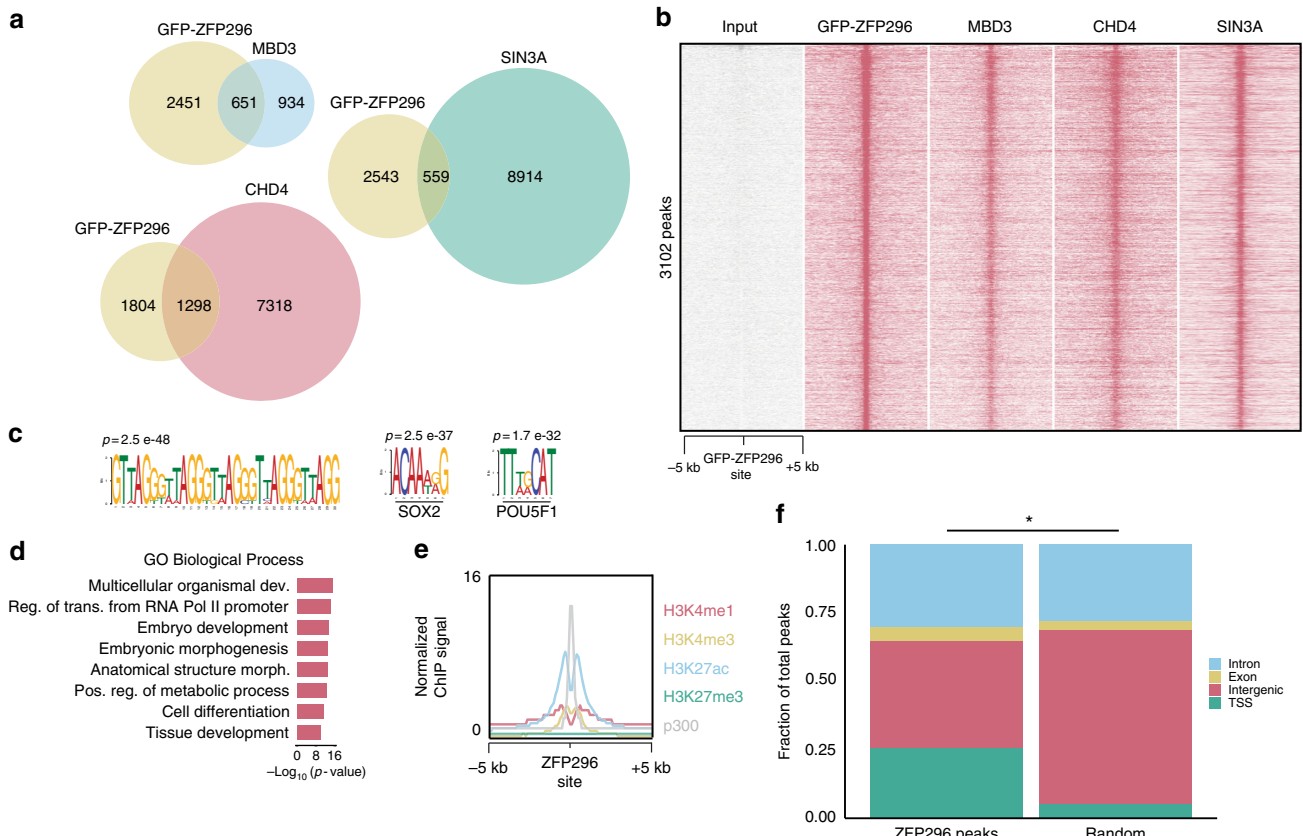

**Fig. 3** ZFP296 co-localizes with NuRD and SIN3A in ESCs. **a** Venn diagrams summarizing the overlap between GFP-ZFP296 peaks and NuRD or SIN3A ChIP-seq results in ESCs. **b** Heat map showing the ChIP-seq read density for input, GFP-ZFP296, MBD3, CHD4, and SIN3A, centred on GFP-ZFP296 ChIP-seq peaks. **c** Top three most significant motifs enriched under GFP-ZFP296 ChIP-seq peaks. **d** GO term enrichment analysis of Biological Processes for genes within 100 kb of a GFP-ZFP296 peak (nearest gene only). **e** Band plot of four different histone marks and p300 (publicly available data[19]) at GFP-ZFP296 binding sites. **f** Genomic distribution of GFP-ZFP296 ChIP-seq peaks in ESC compared to a set of random regions ($n = 3102$) of the same average length. TSS peaks were +5 kb/−1 kb from a transcription start site. *$p < 0.05$, as assessed by Chi-squared test

(Fig. 3a, b). Strikingly, the most enriched DNA sequence motif under ZFP296 peaks is TTAGGG, which is the telomere repeat motif (Fig. 3c). Additionally, the SOX2 and POU5F1/OCT4 DNA binding motifs are also enriched at GFP-ZFP296 binding sites (Fig. 3c). Furthermore, ZFP296 target genes (Supplementary Data 1) are enriched for Gene Ontology (GO) terms related to embryonic development (Fig. 3d). Lastly, we found that ZFP296 binding sites in the mouse ESC genome are marked with H3K27ac, H3K4me1, H3K4me3, and p300[19] (Fig. 3e). All these findings are in agreement with the NuRD ChIP-seq results (Fig. 1e and Supplementary Fig. 1c and d). ZFP296 binding sites map to a mix of TSS (26%), intergenic (39%) and intronic regions (30%), which is slightly enriched for TSS when compared to the genomic distribution of a random subset of the same size and average length (Fig. 3f). These results indicate that, similar to the core NuRD subunits, most GFP-ZFP296 binding occurs at active promoters and enhancers in ESCs.

**KO of *Zfp296* decreases NuRD binding genome-wide.** Since ZFP296 is both a putative DNA binding protein and has a stem cell-specific expression pattern, we hypothesized that ZFP296 may be recruiting the NuRD complex to the ESC-specific loci identified in Fig. 1. To address this hypothesis, we generated several *Zfp296* KO ESC clones using CRISPR/Cas9 and validated them by mass spectrometry (Fig. 4a and Supplementary Fig. 3a, b). Next, we performed ChIP-seq for MBD3 and CHD4 in two of these *Zfp296* KO cell lines while using a spike-in method to allow for relative quantification of the detected ChIP signal in wild-type

versus KO cells[26]. Deletion of ZFP296 from mouse ESCs resulted in a decrease in NuRD binding particularly at ESC-specific NuRD binding sites (Fig. 4b–d and Supplementary Fig. 3c, d), suggesting that ZFP296 could contribute to the recruitment of NuRD to these loci in a stem cell-specific manner. Indeed, loss of MBD3 at ESC-enriched MBD3 binding sites in *Zfp296* KO ESCs is more pronounced when there is co-localization with ZFP296, supporting such a recruiting function (Fig. 4e and Supplementary Fig. 3e). Furthermore, a modest but reproducible average decrease in NuRD binding could also be observed when looking at all ZFP296 binding sites in ESCs (Fig. 4f and Supplementary Fig. 3f, g). Importantly, these changes are unlikely to be caused by changes in protein abundance, as NuRD complex subunits are not significantly altered in *Zfp296* KO versus WT ESCs (Fig. 4a and Supplementary Fig. 3a, b). Since we have shown that ZFP296, apart from NuRD, also interacts and co-localizes with SIN3A, we performed spike-in ChIP-seq for SIN3A in *Zfp296* KO and WT ESCs as well. However, SIN3A levels at ZFP296 sites were not significantly altered in both *Zfp296* KO cell lines compared to WT levels (Supplementary Fig. 3h). While the variance between the two KO lines may be caused by a difference in ZFP296 depletion (Fig. 4a and Supplementary Fig. 3a), these findings might also suggest that the possible recruitment function of ZFP296 could be more specific for NuRD. Lastly, we also used spike-in ChIP-seq to study H3K27ac levels at ZFP296 sites in *Zfp296* KO versus WT ESCs, since H3K27ac is a known substrate for the NuRD complex[27], which showed that H3K27ac levels decreased in the one but increased in the other *Zfp296* KO cell line compared to WT

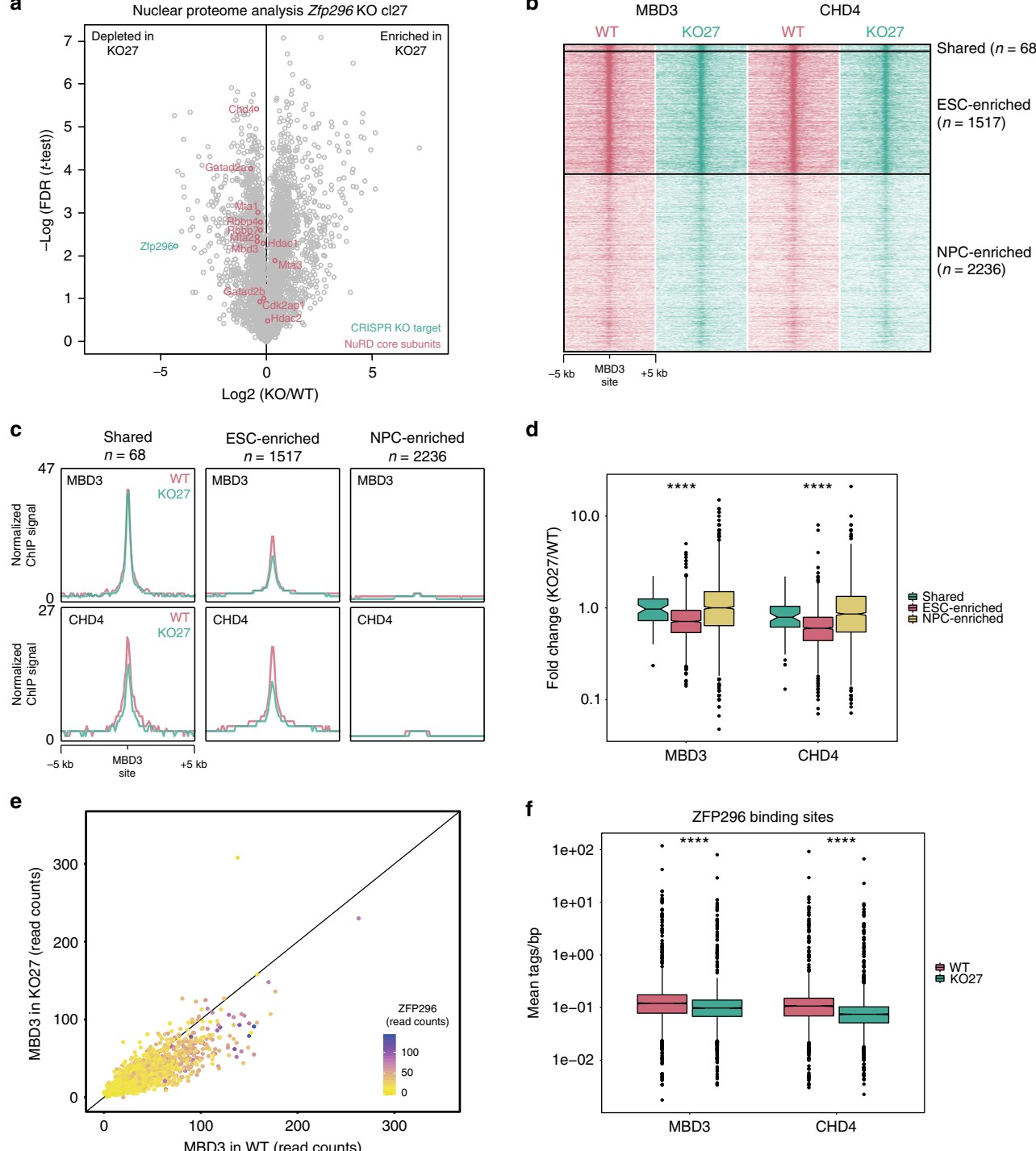

**Fig. 4** ZFP296 knockout decreases NuRD binding genome-wide. **a** Volcano plot from label-free whole proteome mass spectrometry analysis of WT and *Zfp296* KO clone 27 ESCs, graphed as in Fig. 2a. The CRISPR KO target *Zfp296* is colour-coded in green, NuRD core subunits are colour-coded in red. **b** Heat map showing the ChIP-seq read density for MBD3 and CHD4 in WT and *Zfp296* KO clone 27 ESCs, centred on the union of MBD3 peaks from ESC and NPC. **c** Band plots of MBD3 and CHD4 ChIP-seq in WT and *Zfp296* KO clone 27 ESCs at MBD3 binding sites for each class of NuRD binding. **d** Fold change in ChIP-seq read counts (*Zfp296* KO clone 27/WT) of MBD3 and CHD4 at MBD3 binding sites for each class of NuRD binding. Box: median (central line), first and third quartile (box limits); whiskers: 1.5 × interquartile range. ****$p < 0.0001$, as assessed by ANOVA. **e** Scatterplot of MBD3 ChIP-seq read counts at ESC-enriched MBD3 binding sites in WT and *Zfp296* KO clone 27 ESCs, coloured for the amount of ZFP296 at the same locus. The diagonal line indicates no change between the two conditions. A linear model on these data performs significantly better when taking into account ZFP296 levels as a predictive variable ($p < 0.001$, as assessed by ANOVA). **f** Boxplots of average MBD3 and CHD4 ChIP-seq signal in WT and *Zfp296* KO clone 27 ESCs at all GFP-ZFP296 peaks. Box: median (central line), first and third quartile (box limits); whiskers: 1.5 × interquartile range. ****$p < 0.0001$, as assessed by Kruskal–Wallis test. See also Supplementary Fig. 3

ESCs (Supplementary Fig. 3h). Although also here the difference in ZFP296 depletion may explain some of the variance between the two lines, these findings may be in line with recent data showing H3K27ac levels only change very transiently after NuRD binding[13]. Together, these results indicate that ZFP296 may contribute to ESC-specific NuRD binding.

**Zfp296 KO cells exhibit delayed differentiation**. To investigate the global effects of the observed decrease in NuRD binding genome-wide in Zfp296 KO versus wild-type ESCs, we performed RNA sequencing and quantitative whole proteome analyses (Fig. 5a, b). Despite the observed genome-wide changes in NuRD binding, only mild effects on the global transcriptome and proteome were observed. In total, 255 genes (out of 11,151 detected, 2.3%) were significantly regulated at the transcript level and 134 proteins (out of 4780 detected, 2.8%) were differentially expressed between Zfp296 KO and wild-type ESCs (Supplementary Data 3). Of the genes whose transcript expression significantly changed, 17% were bound by MBD3, and 23% were bound by ZFP296 in wild-type cells, suggesting that these changes are both direct and indirect effects of the KO. We observed a roughly equal proportion of genes up- and downregulated in KO versus wild-type cells. Although we only observed mild changes in gene expression, a few interesting genes were significantly downregulated in Zfp296 KO cells at the transcript and protein level. We verified two of these, Dazl and Lefty2, using qRT-PCR analysis (Fig. 5c). Based on the downregulation of these genes, which are important for early lineage commitment, we hypothesized that Zfp296 KO cells may display impaired differentiation capacity. To investigate this, we differentiated Zfp296 KO and empty vector (EV) ESCs into embryoid bodies. We followed a 4-day time course after LIF withdrawal from the culture medium and observed a significant delay in the upregulation of several lineage identity genes across all three germ layers (Fig. 5d). Pluripotency-associated genes were efficiently downregulated in both cell lines. Thus, Zfp296 KO cells are pluripotent but are impaired in their ability to switch on lineage specification genes.

## Discussion

Here, we have identified the zinc finger protein ZFP296 as an embryonic stem cell-specific interactor of the NuRD complex, which additionally interacts with the SIN3A complex as well. This shared interaction may be explained by the fact that NuRD and SIN3A are both members of the HDAC1/2 complexes family and as such contain shared subunits (apart from HDAC1/2 also RBBP4/7). However, these subunits are also shared with for example the CoREST complex, which we did not identify to be co-purified with ZFP296 in ZFP296-GFP affinity purifications. It would therefore be interesting to perform direct interaction assays such as cross-linking immunoprecipitation-MS[28] to study which proteins ZFP296 uses to associate with its interaction partners, and on which factors these interactions are dependent. The putative DNA binding ability of ZFP296 suggested a possible function in recruitment of NuRD and/or SIN3A to specific target genes, and indeed ChIP-sequencing experiments showed decreased genome-wide binding of NuRD, but not SIN3A, upon KO of Zfp296. The molecular mechanisms underlying this intriguing observation remain to be elucidated, but could be due to the relative abundance of ZFP296 compared to core NuRD and SIN3A subunits. Future studies focusing on SIN3A complex composition and stoichiometry in ESCs could provide further insights into this.

Motif enrichment analysis revealed a SOX2 and POU5F1/OCT4 binding motif being enriched under NuRD peaks in ESCs, consistent with a previously reported link between the NuRD complex and the pluripotency network[14,29]. Although several studies have reported direct protein-protein interactions between NuRD and pluripotency factors such as POU5F1/OCT4 and SOX2[15,16] we do not observe these or other pluripotency factors as direct interactors of the NuRD complex in MBD3-GFP ESC affinity purifications. Furthermore, we failed to detect NuRD subunits in POU5F1/OCT4 affinity purifications. Thus, even though the NuRD complex binds to genomic loci that are enriched for POU5F1/OCT4 and SOX2 DNA binding motifs in mouse ESCs, the molecular mechanisms responsible for NuRD recruitment to these loci remain to be elucidated.

We and others[5,23,24,29,30] have identified a large number of putative DNA binding substoichiometric NuRD interactors. The N-terminus of ZFP296 carries a motif, RRK, which is conserved in several of these NuRD-interacting zinc finger proteins, such as FOG1 and ZNF827. Interestingly, ZNF827 was recently shown to recruit NuRD to telomeres[31], suggesting that perhaps more RKK-carrying zinc finger proteins regulate NuRD binding at repetitive regions. Indeed, we identified the telomere repeat to be significantly enriched under ZFP296 binding sites in ESCs, although we found no evidence that ZFP296 plays a role in recruiting NuRD to repeats. However, a recent report linked ZFP296 to regulation of H3K9me3 at major satellite repeats in early mouse embryos[32], indicating that the interplay between NuRD-interacting zinc finger proteins and repeat regions remains an area of active interest. Biochemical experiments using recombinant proteins may shed more light on the DNA-binding properties of NuRD-interacting proteins such as ZFP296.

Recent work from the Schoeler lab revealed that ZFP296 stimulates OSKM mediated iPSC formation[25]. In our hands, Zfp296 KO ESCs remain pluripotent but are delayed in their ability to differentiate upon LIF withdrawal from the culture medium. This is in agreement with a recent study that showed that ZFP296 is important for germ cell specification[33]. Additional investigations regarding NuRD and its role in regulating iPSC formation and pluripotency have reported conflicting observations. Work from the Silva and Hendrich labs revealed that NuRD is required for iPSC formation in a context-dependent manner and that increased NuRD abundance can enhance reprogramming efficiency[34]. In contrast, the Hanna lab showed highly efficient and deterministic iPSC formation in the absence of MBD3[35], as well as other NuRD subunits[36]. Further work is required to explore these contrasting results, but it will be interesting to investigate whether ZFP296 also plays a role in NuRD- or SIN3A-regulated iPSC reprogramming.

## Methods

**Cell culture and embryoid body differentiation**. R1 mouse ESCs were obtained from the ATCC and cultured on gelatine-coated plates in DMEM (Gibco) supplemented with 15% HyClone foetal bovine serum (GE Healthcare Life Sciences), GlutaMAX (Gibco), non-essential amino acids (Lonza), sodium pyruvate (Gibco), penicillin–streptomycin (Gibco), β-mercaptoethanol, home-made LIF, 3 μM PD0325901, and 1 μM CHIR99021. NPCs were differentiated and propagated following the protocol from Conti et al.[17] Briefly, ESCs were differentiated into NPCs using DMEM/F12 (Gibco), supplemented with Neurobasal medium (Gibco), N2 and B27 supplements (Gibco), and β-mercaptoethanol. NPCs were maintained in NSA (Euromed) supplemented with GlutaMAX (Gibco), N2 supplement (Gibco), 10 ng/ml bFGF (100-18C, Perprotech), and 10 ng/ml EGF (236-EG, R&D Systems). All cell lines have been tested for mycoplasma contamination.

BACs were tagged according to the protocol from Poser et al.[21] GFP-tagged BAC lines were prepped on NucleoBond BAC 100 columns (Macherey-Nagel) and transfected into ESCs using Lipofectamine LTX Plus (Invitrogen), followed by G418 selection for 10–12 days. Individual colonies were picked, expanded, and screened for GFP expression.

GFP-ZFP296 ESCs were generated by transfection of a GFP-tagged ZFP296 construct into KH2 ESCs[37]. Full-length ZFP296 protein was cloned from mouse complement DNA into the pcDNA3.1 vector (Invitrogen).

The CRISPR/Cas9 system was used to generate a Zfp296 KO cell line. A guide RNA targeting the first exon of Zfp296 (CCTCGCCGCGTAGATCCCGATAC or CCATATCGGATGTGAAGCGTCAA) was cloned into pSpCas9(BB)-2A-Puro

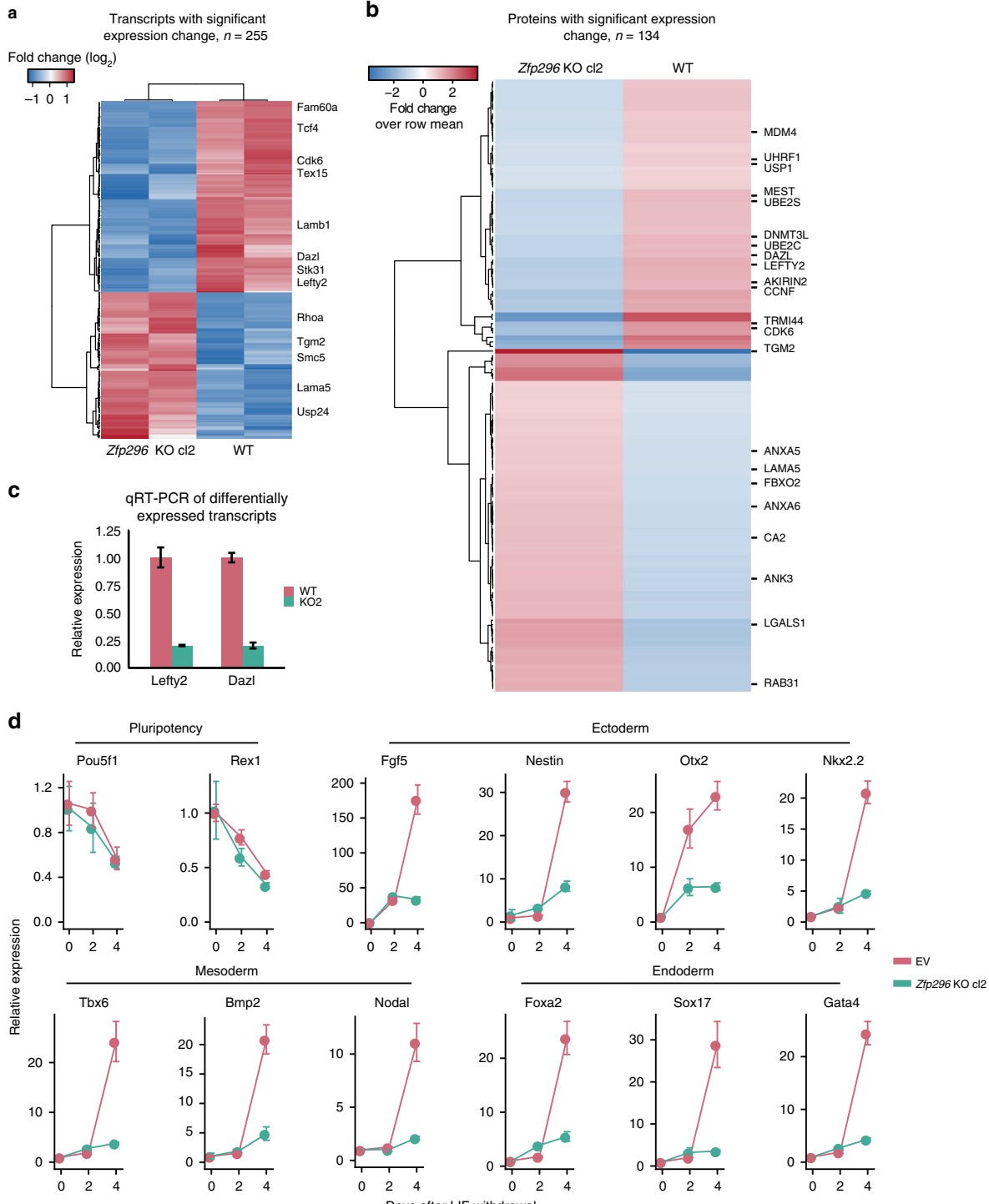

**Fig. 5** *Zfp296* knockout does not affect steady-state gene expression but impairs ESC differentiation. **a** Hierarchical clustering of differentially expressed genes (padj < 0.05) in *Zfp296* KO clone 2 ESCs compared to wild-type (WT) ESCs. **b** Hierarchical clustering of differentially expressed proteins in *Zfp296* KO clone 2 ESCs compared to WT ESCs. **c** qRT-PCR validation of differentially expressed transcripts in WT and *Zfp296* KO clone 2 ESCs. *n* = 2 RNA isolations, error bars = s.d. of technical triplicates from one experiment. **d** qRT-PCR analysis of expression levels of pluripotency and differentiation markers during an embryoid body (LIF withdrawal) time course using WT and *Zfp296* KO clone 2 ESCs. *n* = 4 time courses, error bars = s.d. of technical triplicates from one experiment

(PX459; Addgene 48139). The resulting plasmid was verified by sequencing and transfected into R1 ESCs. After two days, the cells were subjected to puromycin selection for 48 h, then monoclonal cell lines were generated and analyzed by sequencing.

Wild-type and *Zfp296* KO ESCs were differentiated into embryoid bodies by plating 2 × 10⁵ cells/mL of media onto non-adherent plates in the absence of LIF. The medium was changed daily during the differentiation to embryoid bodies.

**Chromatin preparation**. Attached ESCs and NPCs were cross-linked with 1% formaldehyde for 10 min at room temperature with gentle shaking. Cross-linking was quenched with the addition of 1/10 volume 1.25 M glycine. Cells were washed with PBS, then harvested by scraping in Buffer B (20 mM HEPES, 0.25% Triton X-100, 10 mM EDTA, and 0.5 mM EGTA). Cells were pelleted by spinning at 600 × *g* for 5 min at 4 °C. Cell pellet was resuspended in Buffer C (150 mM NaCl, 50 mM HEPES, 1 mM EDTA, and 0.5 mM EGTA) and rotated for 10 min at 4 °C. Cells

were pelleted by spinning at $600 \times g$ for 5 min at 4 °C. The cell pellet was resuspended in 1× incubation buffer (0.15% SDS, 1% Triton X-100, 150 mM NaCl, 1 mM EDTA, 0.5 mM EGTA, and 20 mM HEPES) at 15 million cells/mL. Cells were sheared in a Bioruptor Pico sonicator (Diagenode) at 4 °C using 5 or 7 cycles of 30 s ON, 30 s OFF for ESCs and NPCs, respectively. Sonicated material was spun at $18,000 \times g$ for 10 min at 4 °C, then aliquoted and stored at −80 °C.

**Chromatin immunoprecipitation**. A total of 0.5–10 million cells were used as input for library prep, and 5 million cells were used as input for ChIP-qPCR experiments. Chromatin was incubated overnight together with 1 µg antibody at 4 °C in 1× incubation buffer (0.15% SDS, 1% Triton X-100, 150 mM NaCl, 1 mM EDTA, 0.5 mM EGTA, and 20 mM HEPES) supplemented with protease inhibitors and 0.1% BSA. For ChIPs with spike-in, 25 µg of sample chromatin and 50 or 20 ng of spike-in chromatin (Active Motif) were used for histone modification or transcription factor ChIP, respectively. This chromatin mix was incubated overnight as above, with 2 µg spike-in antibody (Active Motif) and 1 or 3 µg of the antibody of interest for histone or transcription factor ChIP, respectively. A 50/50 mix of Protein A and G Dynabeads (Invitrogen) was added the following day followed by a 90-min incubation. The beads were washed 2× with Wash Buffer 1 (0.1% SDS, 0.1% sodium deoxycholate, 1% Triton, 150 mM NaCl, 1 mM EDTA, 0.5 mM EGTA, and 20 mM HEPES), 1× with wash buffer 2 (wash buffer 1 with 500 mM NaCl), 1× with wash buffer 3 (250 mM LiCl, 0.5% sodium deoxycholate, 0.5% NP-40, 1 mM EDTA, 0.5 mM EGTA, and 20 mM HEPES), and 2× with wash buffer 4 (1 mM EDTA, 0.5 mM EGTA, and 20 mM HEPES). After washing, beads were rotated for 20 min at room temperature in elution buffer (1% SDS, 0.1 M NaHCO$_3$). The supernatant was decrosslinked with 200 mM NaCl and 100 µg/mL Proteinase K for 4 h at 65 °C. Decrosslinked DNA was purified using MinElute PCR Purification columns (Qiagen). DNA amount was quantitated using Qubit fluorometric quantitation (Thermo Fisher Scientific). qPCR analysis of ChIP DNA was performed using iQ SYBR Green Supermix (Bio-Rad) on a CFX96 Real-Time System C1000 Thermal Cycler (Bio-Rad). Primers used for qPCR analysis are listed in Supplementary Table 1.

**Illumina high-throughput sequencing and data analysis**. ChIP-seq libraries were prepared using the Kapa Hyper Prep Kit for Illumina sequencing (Kapa Biosystems) according to the manufacturer's protocol with the following modifications. 5 ng ChIP DNA was used as input, with NEXTflex adapters (Bioo Scientific) and 10 cycles of PCR amplification. Post-amplification clean-up was performed with QIAquick MinElute columns (Qiagen) and size selection was done with an E-gel (300 bp fragments) (Thermo Fisher Scientific). Size-selected samples were analyzed for purity using a High Sensitivity DNA Chip on a Bioanalyzer 2100 system (Agilent). Samples were sequenced on an Illumina HiSeq2000 or NextSeq500. The 43 or 75 bp tags were mapped to the reference mouse genome mm9 (NCBI build 37) or *Drosophila* genome dm3 (for spike-in) using the Burrows-Wheeler Alignment tool (BWA) allowing one mismatch[38]. Only uniquely mapped reads were used for data analysis and visualization. Mapped reads were filtered for quality and duplicates were removed. Peak-calling was performed with the MACS 2.0 tool against a reference input sample from the same cell line[39]. Heat maps and band plots were performed using the Python package fluff[40]. ChIP-seq datasets used for generating heat maps and average profiles were normalized for the spike-in, or else for RPKM. Motif analysis was performed using MEME ChIP[41] and Gimme Motifs[42]. GREAT[43] was used for GO term analysis, and *P*-values were computed using a hypergeometric distribution with FDR correction. R was used to generate most of the graphs.

**Nuclear extracts and nuclear pellet solubilization**. Nuclear extracts were prepared essentially according to Dignam et al.[44]. Briefly, cells were harvested with trypsin, washed twice with PBS, and centrifuged at $400 \times g$ for 5 min at 4 °C. Cells were swelled for 10 min at 4 °C in five volumes of Buffer A (10 mM HEPES/KOH, pH 7.9, 1.5 mM MgCl$_2$, 10 mM KCl), and then pelleted at $400 \times g$ for 5 min at 4 °C. Cells were resuspended in two volumes of Buffer A plus protease inhibitors and 0.15% NP-40 and transferred to a Dounce homogenizer. After 30–40 strokes with a Type B pestle, the lysates were spun at $3200 \times g$ for 15 min at 4 °C. The nuclear pellet was washed once with PBS, and spun at $3200 \times g$ for 5 min at 4 °C. The nuclear pellet was resuspended in 2 volumes Buffer C (420 mM NaCl, 20 mM HEPES/KOH, pH 7.9, 20% v/v glycerol, 2 mM MgCl$_2$, 0.2 mM EDTA) with 0.1% NP-40, protease inhibitors, and 0.5 mM dithiothreitol (DTT). The suspension was incubated with rotation for 1 h at 4 °C, and then spun at $18,000 \times g$ for 15 min at 4 °C. The supernatant was aliquoted and stored at −80 °C until further use.

The nuclear pellets remaining after nuclear extraction were solubilized by resuspension in four volumes of RIPA buffer (150 mM NaCl, 50 mM Tris pH 8.0, 1% NP-40, 5 mM MgCl$_2$, 10% glycerol) plus benzonase (Millipore) at 1000 U/100 ul nuclear pellet. Samples were incubated at 37 °C with shaking until solubilized, then spun at $14000 \times g$ for 5 min at 4 °C. The supernatant was aliquoted and stored at −80 °C until further use.

**Label-free pulldowns**. Label-free GFP pulldowns were performed in triplicate as previously described[45] with the following modifications. For GFP pulldowns, 2 mg of nuclear extract was incubated with 7.5 µl GFP-Trap beads (Chromotek) and

50 µg/mL ethidium bromide in Buffer C (300 mM NaCl, 20 mM HEPES/KOH, pH 7.9, 20% v/v glycerol, 2 mM MgCl$_2$, 0.2 mM EDTA) with 0.1% NP-40, protease inhibitors, and 0.5 mM DTT in a total volume of 400 µl. After incubation, 6 washes were performed: 2 with Buffer C and 0.5% NP-40, 2 with PBS and 0.5% NP-40, and 2 with PBS. Affinity purified proteins were subject to on-bead trypsin digestion as previously described[22]. In short, beads were resuspended in 50 µl elution buffer (2 M urea, 50 mM Tris pH 7.5, 10 mM DTT) and incubated for 20 min in a thermoshaker at 1400 rpm at room temperature. After addition of 50 mM iodoacetamide (IAA), beads were incubated for 10 min at 1400 rpm at room temperature in the dark. Proteins were then on-bead digested into tryptic peptides by addition of 0.25 µg trypsin and subsequent incubation for 2 h at 1400 rpm at room temperature. The supernatant was transferred to new tubes and further digested overnight at room temperature with an additional 0.1 µg of trypsin. Tryptic peptides were acidified and desalted using StageTips[46] prior to mass spectrometry analyses.

**Label-free quantification (LFQ) LC-MS/MS analysis**. Tryptic peptides were separated with an Easy-nLC 1000 (Thermo Scientific). Buffer A was 0.1% formic acid and Buffer B was 80% acetonitrile and 0.1% formic acid. MBD3-GFP ESC and NPC nuclear extract LFQ samples were separated using a 120-min gradient from 7% until 32% Buffer B followed by step-wise increases up to 95% Buffer B. Mass spectra were recorded on a LTQ-Orbitrap Velos mass spectrometer or on a LTQ-Orbitrap Q-Exactive mass spectrometer (Thermo Fisher Scientific), selecting the 10–15 most intense precursor ions of every full scan for fragmentation. The tryptic peptides from GFP-ZFP296 ESC nuclear extracts, *Zfp296* KO ESC nuclear extracts, and ESC nuclear pellet pulldowns were measured by developing a gradient from 9–32% Buffer B for 114 min before washes at 50% then 95% Buffer B, for 140 min of total data collection time. Mass spectra were recorded on an LTQ-Orbitrap Fusion Tribrid mass spectrometer (Thermo Fisher Scientific). Scans were collected in data-dependent top speed mode with dynamic exclusion set at 60 s.

**Label-free and dimethyl-labeled proteomes**. For label-free nuclear proteomes, 100 µg of nuclear extracts were digested using filter-aided sample preparation (FASP)[47] using a 30 kDa cut-off filter and trypsin digest in 50 mM ABC buffer. For dimethyl-labelled whole proteomes, 100 µg of whole cell lysates were digested using FASP using a 30 kDa cut-off filter and trypsin digest in TEAB buffer. For labeling, each sample was differentially labeled after FASP by incorporation of stable isotopes on the peptide level using light and medium dimethyl labeling[48]. Differentially dimethyl-labeled samples were mixed and fractionated by strong anion exchange (SAX)[49]. We collected the flow through (FT) and pH 11, pH 8, pH 5, and pH 2 elutions. The peptides were subjected to Stage-Tip desalting[46] prior to mass spectrometry analysis. Peptides were separated by online nanoLC-MS/MS using a 214 min gradient of acetonitrile (7% to 30%) followed by washes at 60% then 95% acetonitrile. Data were collected on a Fusion Tribrid mass spectrometer for 240 min of total data acquisition time.

**Mass spectrometry data analysis**. Thermo RAW files were analyzed with MaxQuant version 1.5.1.0 or 1.6.0.1 using default settings and searching against the UniProt mouse proteome, release 2015_12 or 2017_06[50]. Additional options for Match between runs, LFQ, and iBAQ were selected where appropriate. Stoichiometry calculations and volcano plots were produced essentially as described[22] using Perseus[51] version 1.4.0.8 and in-house R scripts. Statistical cut-offs were chosen such that no proteins were present as outliers on the control, non-GFP side of the volcano plot.

**RNA-seq sample prep and analysis**. RNA was isolated in duplicate from cells using an RNeasy Mini Kit (Qiagen). Ribosomal RNA was depleted by treatment with the Ribo-Zero rRNA Removal Kit (Illumina) and fragmented into approximately 200 bp fragments in fragmentation buffer (200 mM Tris-acetate, 500 mM KCH$_3$COO, 150 mM Mg(CH$_3$COO)$_2$, pH 8.2). Strand-specific libraries of cDNA were prepared using SuperScript III Reverse Transcriptase (Invitrogen) and a Kapa Hyper Prep Kit, as described above for ChIP-seq, but including an additional incubation with USER enzyme (NEB) before library amplification to digest the second cDNA strand. Reads were mapped onto the reference mouse genome mm9 using hisat[52]. Count tables were generated using HTSeq[53]. Differential gene expression was analysed with the DESeq2 R package.[54]

**Quantitative reverse transcriptase PCR**. RNA was isolated using the RNeasy Mini Kit (Qiagen) and 1 µg of RNA was used for cDNA synthesis with the iScript cDNA Synthesis Kit (Bio-Rad). qRT-PCR was performed using iQ SYBR Green Supermix (Bio-Rad) on a CFX96 Real-Time System C1000 Thermal Cycler (Bio-Rad). Primers used for qRT-PCR analysis are listed in Supplementary Table 2. Gapdh and β-actin were used as the reference genes.

**Co-immunoprecipitation and immunoblotting**. For endogenous immunoprecipitation, 250 µg of nuclear extract in a total volume of 200 µl buffer C (300 mM NaCl, 20 mM HEPES KOH pH 7.9, 20% (v/v) glycerol, 2 mM MgCl$_2$, 0.2 mM EDTA, 0.5% NP-40, 0.5 mM DTT, complete protease inhibitors) was incubated

overnight with anti-MBD3 (IBL, JP10281, 2 μg per IP), followed by incubation with 20 μl of a 1:1 mixture of Protein A and G Dynabeads (Thermo Fisher) at 4 °C for 90 min. For GFP co-IPs, 7.5 μl of GFP-trap agarose beads (Chromotek) were incubated with 250 μg of nuclear extracts in a total volume of 200 μl buffer C for 90 min at 4 °C. The beads were then washed three times with 1 mL buffer C and finally boiled in Laemmli buffer.

Nuclear extracts or input samples (25 μg nuclear extract boiled in Laemmli buffer) or immunoprecipitated proteins were fractionated by SDS-PAGE and transferred to a nitrocellulose membrane using a transfer apparatus according to the manufacturer's protocol (Bio-Rad). After 1 h blocking with 5% milk in TBST (10 mM Tris, pH 8.0, 150 mM NaCl, 0.5% Tween 20) at room temperature, the membrane was incubated overnight at 4 °C using anti-MBD3 (IBL, JP10281, 1:1000 dilution), anti-SALL4 (abcam, ab29112, 1:5000 dilution), or anti-GFP (Roche, 11814460001, 1:2000 dilution). The membrane was washed 3 times with TBST followed by incubation with a 1:3000 dilution of horseradish peroxidase-conjugated anti-mouse or anti-rabbit antibodies (Dako; catalogue number p0260 and p0399, respectively) in 5% milk in TBST at room temperature. Following secondary antibody staining, the membrane was washed 3 times in TBST, followed by development using the ECL Western Blotting Substrate (Promega) and imaging on a ImageQuant LAS4000 (GE Healthcare). Uncropped western blots can be found in Supplementary Fig. 4.

## Data availability

The mass spectrometry proteomics data have been deposited to the ProteomeXchange Consortium via the PRIDE partner repository[55] with the dataset identifier PXD010512. High-throughput sequencing data have been deposited in the GEO database repository with the dataset identifier GSE117289. All figures have associated raw data. All other relevant data are available from the corresponding author upon reasonable request.

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

## Acknowledgements

We are grateful to I. Poser (MPI-CBG Dresden) for contributing the MBD3 BAC. We thank the Sequencing and Bioinformatics teams of the RIMLS Mol(Dev)Bio departments for ChIP-and RNA-seq support. The Vermeulen lab is supported by the EU FP7 framework program 277899 (4DCellFate), an ERC Starting Grant (309384), and the NWO Gravitation program Cancer Genomics Netherlands. The Vermeulen lab is part of the Oncode Institute, which is partly funded by the Dutch Cancer Society (KWF).

## Author contributions

S.L.K. and M.V. designed the study. S.L.K., I.D.K., L.v.V., M.P.B., R.R.E., D.P.L., and P.W.T.C.J. performed experiments. S.L.K., I.D.K., and R.G.L. analyzed data. S.L.K., I.D.K., and M.V. wrote the manuscript together with input from all authors.

## Additional information

**Competing interests:** The authors declare no competing interests.

