## [Peer Review File · Nature Communications]

Reviewer #1 (Remarks to the Author):

In this manuscript Kloet and colleagues investigate a novel NuRD interacting protein, Zfp296, and its role in genome-wide NuRD localization and differentiation of mESCs. The authors compare ChIP-seq patterns of the NuRD subunits Mbd3 and CHD4 in ESCs and neural progenitor cells derived from them. They find that most NuRD binding sites change during differentiation. To identify candidate factors involved in the dynamic redistribution of NuRD binding they affinity purify MBD3 interactors from ESCs and NPCs. They identify the zinc finger protein Zfp296 as a strong NuRD interactor in ESCs. However, this interaction is not detected in NPCs. ChIP-seq analysis of an epitope-tagged Zfp296 protein demonstrates widespread overlap with NuRD binding sites in ESCs. The authors analyse histone modifications enriched at Zfp296/NuRD binding sites and report an enrichment of H3K27ac and H3K4me1. Therefore, they conclude that most binding sites map to active enhancers. Surprisingly, KO of Zfp296 in ESCs does not result in a decrease of NuRD chromatin binding. Rather, CHD4 and MBD3 binding is globally increased concomitant with increased H3K27 acetylation. RNA-seq analysis of ESC Zfp296 KO cells reveals a modest change in the transcriptome. Nevertheless, several differentiation-specific genes are downregulated in the course of differentiation. In agreement with this finding Zfp296 KO cells display delayed differentiation kinetics.

The role of ATP-dependent chromatin remodelers in ESC cell biology is of great interest. An outstanding question in the field is how these developmental regulators are directed to chromatin and how they modulate expression of different sets of target genes in the course of differentiation. As such, this manuscript is timely and has the potential to make an important contribution. The manuscript is very well written and the data are generally of very high quality.

I have one only major concern:

It is surprising that Zfp296 KO results in increased NuRD chromatin binding AND increased K27 acetylation at these sites. Why H3K27ac would increase in the absence of Zfp296 is not explained. Furthermore, the authors imply that an increase in K27ac might contribute to stronger NuRD binding (line 179). I do not find this convincing: Would NuRD binding not decrease K27ac due to its HDAC activity? A possible solution to this conundrum would be if other HDAC complexes (or HAT complexes) were involved. Indeed, Zfp296 also interacts with the Sin3 HDAC complex (Figure 2). If Sin3 complex binding (unlike NuRD binding) to chromatin depends on Zfp296 this could provide an explanation for increased K27 acetylation. This would be relatively easy to address experimentally, e.g. by analysing Sin3 complex association with regions showing an increase in K27ac (WT vs. Zfp KO).

In addition, Figure 4c convincingly demonstrates the above mentioned increase in NuRD binding by centering on Zfp296 binding sites. By contrast, the (apparently mild) increase in K27ac is shown by centering on K27ac peaks. It would be much more convincing if K27ac would also be centered on Zfp296 or NuRD binding sites to provide a more direct link.

Minor points:

Line 78: "endogenous antibodies" is a strange term. I think the authors mean that they used antibodies to IP endogenous proteins.

To the uninformed reader, there appears to be a discrepancy in the numbers of peaks shared by MBD3 and CHD4 in Figures 1a and 1c. For example, Mbd3 and CHD4 share 1354 peaks in ESCs (Figure 1a). However, in Figure 1c this number appears to be higher (541 plus 1149). Please, explain the reason for this.

It is remarkable that the majority of CHD4 peaks in ESCs and NPCs does not overlap with MBD3 peaks. It would be interesting to know if the CHD4 binding sites that do not change during differentiation are predominantly CHD4-only sites or NuRD sites.

Figure 1e: It is interesting that the distribution of K4me1 and K27ac around MBD3 binding sites shows a different pattern in cluster 2 (two maxima) and cluster 3 (one maximum). What might the underlying molecular mechanism be? The authors should discuss this.

Figure 2c and 2d: The authors state that the "core NuRD complex is quite stable during cellular differentiation" (line 137). However, there appear to be twofold changes in the stoichiometries of Rbbp4/7 and Chd3/4 - does this not imply that the composition of the NuRD complex does change to some extent?

Figure 2e: When GFP-Zfp296 interactors are purified mESCs most NuRD subunits are recovered with good efficiency. However, Chd4 itself does not appear to be recovered significantly (below/at threshold line). Could this mean that Zfp296 preferentially interacts with NuRD sub-complexes not containing the remodeler?

Figure 3g: The accumulation of H3K4me1 around GFP-Zfp296 binding sites does not look convincing. Can the authors compare with additional ChIP-seq data that would strengthen the conclusion that GFP-Zfp295 associates predominantly with active enhancers (e.g. CBP, Mediator, BRD4 etc.)?

Supplementary Figure 3f: This figure demonstrates that NuRD composition does not change. Can the authors label the NuRD subunits in this figure to make this clearer?

Figure 5: It would be interesting to correlate the changes in transcription with changes in NuRD binding upon Zfp296 KO. Are the genes that change expression bound by NuRD and/or Zfp296? If so, does NuRD binding increase at these genes in the KO cells?

Reviewer #2 (Remarks to the Author):

The manuscript “NuRD-interacting protein Zfp296 regulates genome-wide NuRD localization and differentiation of mouse embryonic stem cells” by Kloet et al characterises cell-type specific NuRD complexes in ESCs and NPCs. The authors present ChIP-sequencing data which revealed a striking difference in NuRD binding sites in embryonic stem cells (ESCs) compared to neural progenitor cells (NPCs), with a significant proportion of sites in ESCs located at enhancers. They identify Zfp296 as a novel Mdb3/NuRD-interacting protein in ESCs and reciprocal experiments with tagged-Zfp296 confirms this interaction. A genome-wide increase in NuRD binding and H3K27ac levels is observed in Zfp296-depleted cells, although this had a minimal effect on NuRD target gene transcription and expression. The delayed up-regulation of differentiation genes in Zfp296-KO embryoid bodies supporting their hypothesis that NuRD, and specifically the Zfp296 subunit, may be essential for ESC pluripotency. The manuscript is well written and experiments were carefully executed. The findings have potential interest for wide audience, including the chromatin and embryonic stem cell fields, however there are several points that need to be addressed prior to publication.

Major comments:

1) In Figure 1(e), the authors presented H3K27ac, H3K4me3, H3K27me3 ChIP-seq average read plots from ESC (Cluster 1 and 2) and NPC (Cluster 3). However, they should provide all tracks from all cell types, together.

2) The mass spec data in Figure 2(e) shows that GFP-Zfp296 interacts with several Sin3a-complex proteins, which has been recently shown by Streubel et al to bind to all H3K4me3 positive gene promoters (and not enhancers) in ESCs. The authors should comment on the potential functional interplay between Zfp296 and the Sin3a complex and gene promoters (if they believe there is any) and how this might / might not affect their discussion of its role together with NuRD at enhancers.

3) The ChIP-seq analysis in Figure 4 is not quantitative. ChIP-Rx for H3K27ac should be attempted in wt and KO cells and quantitative ChIP qPCR validations of Mbd3 and ChD4 should be performed at multiple different target sites in WT and KO ESCs.

4) The authors should validate the results in Figure 4 with two independent Zfp296- KO ESC clones.

5) Western blots and/or RT-qPCR of Zfp296 KO cells should be included in Figure 4/Supplemental Figure 3 to confirm the knockout.

Minor comments:

1) The authors should clarify in the text and legend what “Cluster 1, 2 and 3” are in Figure 1.

2) The authors should clarify why Mbd3 (the bait) is presenting as a high fold enrichment in only NPCs and not ESCs in Figure 2(d).

3) The yellow ChIP-seq plots are difficult to see on a print out of the paper.

Reviewer #3 (Remarks to the Author):

I read with great interest the manuscript by Kloet et al. entitled “NuRD-interacting protein Zfp296 regulates genome-wide NuRD localization and differentiation of mouse embryonic stem cells”. In this manuscript the authors use proteomic and genomic approaches to study NuRD complex dynamics during ESC differentiation into NPCs. They uncovered Zfp296 as novel NuRD interacting protein in ESC that seems to globally restrain NuRD binding in ESC and to contribute to proper ESC differentiation.

The role of the NuRD complex in ESC and its importance for pluripotency has been extensively studied. However, the role of NuRD during differentiation, the dynamics of its genomic targets and its cell-type specific interacting partners are much less studied. Therefore, in principle, this manuscript addresses an interesting question. However, in the current format I am afraid I can not support publication in Nature Communications due to some important concerns:

- Although the authors acknowledge that NuRD composition, genomic targets and functional role has been extensively studied in ESC, they still decided to focus on how NuRD function is altered in ESC, in this case in the absence of Zfp296. Why not focusing on some of the NPC specific NuRD interacting proteins?. This could have made the study considerably more novel and original.
- The authors provide very limited mechanistic insights into how Zfp296 controls NuRD genomic targeting and/or enzymatic activity. They describe some potentially interesting observations, which unfortunately do not follow at all: (i) Zfp296 interacts with NuRD in the nucleoplasm rather than on chromatin, which could suggest that Zfp296 controls sub-cellular NuRD localization. Rather than following this idea, they hypothesize that Zfp296 binds to regulatory elements and recruits NuRD to such genomic targets. If this would be the case, then why Zfp296 preferentially interacts with NuRD in the nucleoplasm?. (ii) Based on ChIP-seq for Zfp296, the most overrepresented motif is the telomere repeat, which could be a potentially very interesting observation. However, this is barely mentioned in the rest of the manuscript. Are telomeres frequently bound by Zfp296 and NuRD?. Are they co-bound (sequential ChIPs)?. How is NuRD binding to telomeres affected in Zfp296 KO mESC?. Are telomeres affected in Zfp296 KO mESC and could this affect their pluripotency and differentiation potential?.
- The authors claim that the loss of Zfp296 in ESC globally affects H3K27ac. However, this claim is not totally supported by their data, which is not clearly presented. The differences shown in Fig4D seem rather minor and the shape of the H3K27ac signal is weird, as it does not show the typical dip in the middle. How many replicates did the author use?. Are the presented differences statistically significant?. The authors should present H3K27ac data as heatmaps to appreciate whether all H3K27ac loci are moderately affected or, alternatively, whether a few loci are more severely affected?. Are the Zfp296 bound loci more sensitive to changes in H3K27ac and/or NuRD than other H3K27ac loci?. Last but not least, how do the authors interpret an increase in H3K27ac with a concomitant increase in NuRD binding in the Zfp296 KO ESC?. Are HDAC1/2 also being recruited to these loci in the Zfp296 KO ESC?.
- The characterization of the differentiation defects of Zfp296 KO ESC seems rather superficial, preliminary and correlative?. Are the genes showing differential expression in Zfp296 KO ESC preferentially bound by Zfp296?. Do these genes show preferential changes in H3K27ac and/or NuRD binding in Zfp296 KO ESC?. Their data upon embryoid body differentiation suggests a failure to differentiate: is this due to Zfp296 being directly involved in activation of lineage identity genes or, alternatively, due to a direct role of Zfp296 in the silencing of pluripotency genes?. In this regard, evaluating only Pou5f1 and Rex1 is clearly insufficient.

Additional minor comments:

- In Fig 1E, histone modification signals in each cluster should be shown in both ESC and NPC.

- The ontology terms shown in Supp Fig 1, they do not seem to be as different between the different NuRD binding sites as the authors claim in the text. Developmental terms dominate in all three categories, without an enrichment of pluripotency or neural terms in either the ESC or NPC-specific NuRD binding sites.
- Fig 3A-C. The authors should perform sequential ChIP followed by qPCR analysis to investigate whether Zfp296 and NuRD subunits are simultaneously bound to their shared genomic targets.

Reviewer #4 (Remarks to the Author):

Kloet and colleagues describe in this paper the protein Zfp296 as a new interactor of the NuRD complex in mouse embryonic stem cells. This finding is well supported by different experimental approaches and provide new insights into the function of the protein Zfp296. However, there are several aspects of the manuscript in need of some revision before the manuscript is suitable for publication.

Major Comments

- 1.- The authors should clarify why they assign all the effect on acetylation upon Zfp296 KO to the NuRD complex when Zfp296 is a shared interactor of the NuRD and Sin3/Hdac complexes. They should also perform ChIP on a component of the Sin3/Hdac complex in WT and upon Zfp296 to see the effect in this complex. As a general comment, even if the focus of the manuscript is on the effect of Zfp296 on the NuRD complex, the role of Sin3/HDAC in the story should be addressed in more detail because of the shared Zfp296 interaction.
- 2.- Line 170: The authors should perform ChIP on one or two specific components of the Sin3/HDAC complex to see if distribution of this complex is also affected by the KO of Zfp296 in ESCs or NPCs.
- 3.- Lines 88-90: The sentence "In contrast, cell-type specific NuRD binding sites mostly map to intergenic and intronic regions which are marked with H3K27ac and H3K4me1, indicating that dynamic NuRD binding likely occurs at active enhancers (Fig. 1c-e)" is only applicable to NPC. For the ESC cell (Fig 1 Panel 2), the H3K4me3 is as higher as the H3K4me1.
- 4.- Figure 1e: The authors should also comment on the differences in the distribution of the histone marks relative to the Mbd3 site in ESC and NPC (double peak depleted in the centre for cluster 2 versus single centred peak for cluster 3). Are these findings reproducible when analysing the Chd4 ChIP data the same way as represented in Fig 1 d/e? Is this dual distribution due to the Sox2/Oct4

binding motifs for ESC and Fos/Jun for NPC? Are there other factors described showing this binding pattern?

5.- Lines 173-176: Is the observed difference in Fig4d and e significant? The mentioned increment is around 0.1 mean tags/bp. Is this enough to build the argument that follows this part?

Minor Comments

Line 61-62: Explain the developmental connection between ESC and NPC, or refer it to an existing paper/review.

Line 70: Define LIF

Line 84: Give a number/percentage instead of “many”

Line 88: Give a number/percentage instead of “mostly”

Line 122: Rephrase “Zfp296 is relatively uncharacterized protein”. Describe what is known and what is unknown or unusual to transmit a more defined idea of the knowledge on Zfp296.

Line 136: Which is the stoichiometry of the NuRD and the Sin3/Hdac complexes normalized to the bait in the Zfp296 IP? Could the authors estimate from there which is the fraction of Zfp296 interacting with each complex? This may help to answer the Major Comment 1?

Line 137/138: “the core NuRD complex is quite stable”. Define the stability of the complex using data. In line 45 the proteins Cdk2ap1/Doc1 and Mbd3 are described as part of the NuRD core complex. However, in Fig2d, it seems that Cdk2ap1/Doc1 and Mbd3 are enriched in the NPC when compared to the ESC, as this will point to different composition of the complex depending on the cell state. The authors should comment on that. Plotting the comparison between Mbd3-GFP in mESC and NPC to see the changes of all the NuRD core complex components will help to visualize possible changes in complex composition.

Line 143 “many of these overlap” give a number.

Line 144: “the most enriched DNA sequence motif under Zfp296 peaks is TTAGGG, which is the telomere repeat motif (Fig. 3d).” The authors should mention if there is any known telomere interacting protein enriched in the Zfp296 IP and highlight it in the corresponding plot (Figure 2e).

Line 149: “Zfp296 binding sites map to a mix of transcription start sites (26%), intergenic (39%) and intronic regions (30%)” The authors should compare these percentages with the ones of TSS, intergenic and intronic regions in the complete genomic, to see if the distribution is normal or enriched for any of the three classes.

Line 150/153/161: Give a number for “large majority”, “small percentage” and “a global increase”.

Line 151: Fig3g should also have the same kind of plotting that Fig 1e is representing for the ChIP Mbd3 data.

Line 190: The authors should give both the absolute and the percentage of differentially expressed genes/proteins in the Zfp296 KO vs WT ESC comparison.

Line 206: Comments on the role of Sin3/Hdac.

Line 372: "Statistical cut-offs were chosen such that no proteins were present as outliers on the control, non-GFP side of the volcano plot." Volcano plot Fig2b is not following the rule described here.

Line 407: Submission of the data should be done and PDX number should be added before accepting the manuscript.

Fig1: The number of peaks in the intersection of the Venn diagrams should be the same as the cluster 1 shared from Panel c. Otherwise explain the relationship between the numbers from Panel a and c. Scale of plots from Panel e should be equal to allow better comparison. Why are the numbers in cluster 1 from Panel e so high when only 541 peaks are used, and for clusters 2 and 3 so low (0.75 and 0.2 max) when the number of peaks are 1149 and 1607?

Fig2: The subunits of the Nurd complex in Panels a and b and the subunits of Nurd and Sin3 complexes in Panel e should be colour coded. The original data from the bar chart in panel d should be also displayed as volcano plots to get a better overview of the differences between ESC and NPC changes. In this volcano plot the subunits of Nurd and Sin3 should be colour coded as well.

We would like to thank the reviewers for their careful reading of our manuscript, and are pleased that all reviewers think our results are of high quality and interesting for a wide audience. We are also grateful for the reviewers' constructive feedback on how to improve our study. Based on these suggestions, we have significantly revised our manuscript. Notably, the inclusion of a spike-in ChIP-seq control as suggested by Reviewer 2 led us to conclude that *Zfp296* knockout actually decreases, rather than increases, NuRD binding genome-wide. Below, we provide a point-to-point response to the reviewer comments, where the original comments are in italics and our response is in regular font.

Reviewer #1

In this manuscript Kloet and colleagues investigate a novel NuRD interacting protein, Zfp296, and its role in genome-wide NuRD localization and differentiation of mESCs. The authors compare ChIP-seq patterns of the NuRD subunits Mbd3 and CHD4 in ESCs and neural progenitor cells derived from them. They find that most NuRD binding sites change during differentiation. To identify candidate factors involved in the dynamic redistribution of NuRD binding they affinity purify MBD3 interactors from ESCs and NPCs. They identify the zinc finger protein Zfp296 as a strong NuRD interactor in ESCs. However, this interaction is not detected in NPCs. ChIP-seq analysis of an epitope-tagged Zfp296 protein demonstrates widespread overlap with NuRD binding sites in ESCs. The authors analyse histone modifications enriched at Zfp296/NuRD binding sites and report an enrichment of H3K27ac and H3K4me1. Therefore, they conclude that most binding sites map to active enhancers. Surprisingly, KO of Zfp296 in ESCs does not result in a decrease of NuRD chromatin binding. Rather, CHD4 and MBD3 binding is globally increased concomitant with increased H3K27 acetylation. RNA-seq analysis of ESC Zfp296 KO cells reveals a modest change in the transcriptome. Nevertheless, several differentiation-specific genes are downregulated in the course of differentiation. In agreement with this finding Zfp296 KO cells display delayed differentiation kinetics.

The role of ATP-dependent chromatin remodelers in ESC cell biology is of great interest. An outstanding question in the field is how these developmental regulators are directed to chromatin and how they modulate expression of different sets of target genes in the course of differentiation. As such, this manuscript is timely and has the potential to make an important contribution. The manuscript is very well written and the data are generally of very high quality.

Major points

1. It is surprising that Zfp296 KO results in increased NuRD chromatin binding AND increased K27 acetylation at these sites. Why H3K27ac would increase in the absence of Zfp296 is not explained. Furthermore, the authors imply that an increase in K27ac might contribute to stronger NuRD binding (line 179). I do not find this convincing: Would NuRD binding not decrease K27ac due to its HDAC activity? A possible solution to this conundrum would be if other HDAC complexes (or HAT complexes) were involved. Indeed, Zfp296 also interacts with the Sin3 HDAC complex (Figure 2). If Sin3 complex binding (unlike NuRD binding) to chromatin depends on Zfp296 this could provide an explanation for increased K27 acetylation. This would be relatively easy to address experimentally, e.g. by analysing Sin3 complex association with regions showing an increase in K27ac (WT vs. Zfp KO).

Following the suggestion of another reviewer, we performed ChIP-seq using a spike-in for H3K27ac in WT and *Zfp296* KO ESCs. Based on these analyses, we conclude that the previous change we saw in H3K27ac was not significant; indeed, our new normalised data do not show such a difference. We additionally performed ChIP-seq analysis on SIN3A in WT and KO ESCs, where we didn't see an indication that SIN3A complex binding to chromatin is regulated by ZFP296.

2. In addition, Figure 4c convincingly demonstrates the above mentioned increase in NuRD binding by centering on Zfp296 binding sites. By contrast, the (apparently mild) increase in

K27ac is shown by centering on K27ac peaks. It would be much more convincing if K27ac would also be centered on Zfp296 or NuRD binding sites to provide a more direct link.

We thank the reviewer for this suggestion. Our new band plots showing H3K27ac are now centred on ZFP296 peaks to better evaluate their relationship.

Minor points

3. Line 78: "endogenous antibodies" is a strange term. I think the authors mean that they used antibodies to IP endogenous proteins.

We adjusted this phrase.

4. To the uninformed reader, there appears to be a discrepancy in the numbers of peaks shared by MBD3 and CHD4 in Figures 1a and 1c. For example, Mbd3 and CHD4 share 1354 peaks in ESCs (Figure 1a). However, in Figure 1c this number appears to be higher (541 plus 1149). Please, explain the reason for this.

We thank the reviewer for bringing this to our attention. This discrepancy was caused by us considering only one MBD3 ChIP-seq replicate for the one figure while considering two for the other. We now changed all figures to agree with Fig. 1a which is based on 2 replicates.

5. It is remarkable that the majority of CHD4 peaks in ESCs and NPCs does not overlap with MBD3 peaks. It would be interesting to know if the CHD4 binding sites that do not change during differentiation are predominantly CHD4-only sites or NuRD sites.

Since we know that CHD4 can also act independently of NuRD (see e.g. O'Shaughnessy & Hendrich, 2013, Biochem. Soc. Trans), we therefore think that the CHD4 sites that do not overlap with MBD3 peaks are predominantly CHD4-only sites. This is also in agreement with recent data from MBD3 and CHD4 ChIP-seq (Bornelöv *et al.*, 2018, Mol. Cell). We mention this in the main text now as well.

6. Figure 1e: It is interesting that the distribution of K4me1 and K27ac around MBD3 binding sites shows a different pattern in cluster 2 (two maxima) and cluster 3 (one maximum). What might the underlying molecular mechanism be? The authors should discuss this.

This is interesting indeed, as we mention in the text now, although we could not elucidate any biological mechanism behind this. However, recent data on H3K27ac at NuRD-bound loci also found that a single or double H3K27ac peak is characteristic of promoters or active enhancers, respectively (Bornelöv *et al.*, 2018, Mol. Cell), supporting our conclusions.

7. Figure 2c and 2d: The authors state that the "core NuRD complex is quite stable during cellular differentiation" (line 137). However, there appear to be twofold changes in the stoichiometries of Rbbp4/7 and Chd3/4 - does this not imply that the composition of the NuRD complex does change to some extent?

We thank the reviewer for bringing this to our attention. There are indeed some changes in the composition of the NuRD complex, although this could be an artefact of the BAC system and we have no reason to assume this greatly changes its biology. We have now changed the text to avoid any confusion.

8. Figure 2e: When GFP-Zfp296 interactors are purified mESCs most NuRD subunits are recovered with good efficiency. However, Chd4 itself does not appear to be recovered significantly (below/at threshold line). Could this mean that Zfp296 preferentially interacts with NuRD sub-complexes not containing the remodeler?

The hypothesis suggested by the reviewer is interesting, but we do not have sufficient experimental evidence to address this question at this point (e.g. CHD4 pull-downs). In addition, we do not see an enrichment of CHD3 or CHD5, which could suggest that maybe this is mainly NuRD without any CHD. However, we do see colocalisation of ZFP296 with CHD4 in ChIP-seq experiments. Also technical aspects could play a role; the ZFP296-GFP ESC line is an overexpression line rather than a BAC line, hence the resulting interactions may not reflect the wildtype interactions that well. Furthermore, we know from structural work from our lab and others that CHD4 is a peripheral NuRD subunit that may get washed away in affinity purifications. For now, we did not address this issue further, since it would not change our main conclusions.

9. Figure 3g: The accumulation of H3K4me1 around GFP-Zfp296 binding sites does not look convincing. Can the authors compare with additional ChIP-seq data that would strengthen the conclusion that GFP-Zfp295 associates predominantly with active enhancers (e.g. CBP, Mediator, BRD4 etc.)?

We thank the reviewer for this suggestion. We now presented these data as a band plot, and included p300 data to strengthen our conclusions.

10. Supplementary Figure 3f: This figure demonstrates that NuRD composition does not change. Can the authors label the NuRD subunits in this figure to make this clearer?

We thank the reviewer for this suggestion, and we have changed the figure accordingly (now Supplementary Fig. 3b).

11. Figure 5: It would be interesting to correlate the changes in transcription with changes in NuRD binding upon Zfp296 KO. Are the genes that change expression bound by NuRD and/or Zfp296? If so, does NuRD binding increase at these genes in the KO cells?

We thank the reviewer for this suggestion. We performed the suggested analysis, and found that of the genes whose transcript expression significantly changed upon Zfp296 KO, 17% were bound by MBD3, and 23% were bound by ZFP296 in wild-type cells.

Reviewer #2

The manuscript "NuRD-interacting protein Zfp296 regulates genome-wide NuRD localization and differentiation of mouse embryonic stem cells" by Kloet et al characterises cell-type specific NuRD complexes in ESCs and NPCs. The authors present ChIP-sequencing data which revealed a striking difference in NuRD binding sites in embryonic stem cells (ESCs) compared to neural progenitor cells (NPCs), with a significant proportion of sites in ESCs located at enhancers. They identify Zfp296 as a novel Mdb3/NuRD-interacting protein in ESCs and reciprocal experiments with tagged-Zfp296 confirms this interaction. A genome-wide increase in NuRD binding and H3K27ac levels is observed in Zfp296-depleted cells, although this had a minimal effect on NuRD target gene transcription and expression. The delayed up-regulation of differentiation genes in Zfp296-KO embryoid bodies supporting their hypothesis that NuRD, and specifically the Zfp296 subunit, may be essential for ESC pluripotency. The manuscript is well written and experiments were carefully executed. The findings have potential interest for wide audience, including the chromatin and embryonic stem cell fields, however there are several points that need to be addressed prior to publication.

Major comments

1. In Figure 1(e), the authors presented H3K27ac, H3K4me3, H3K27me3 ChIP-seq average read plots from ESC (Cluster 1 and 2) and NPC (Cluster 3). However, they should provide all tracks from all cell types, together.

We thank the reviewer for this suggestion, and we have changed Fig. 1e accordingly.

2. The mass spec data in Figure 2(e) shows that GFP-Zfp296 interacts with several Sin3a-complex proteins, which has been recently shown by Streubel et al to bind to all H3K4me3 positive gene promoters (and not enhancers) in ESCs. The authors should comment on the potential functional interplay between Zfp296 and the Sin3a complex and gene promoters (if they believe there is any) and how this might / might not affect their discussion of its role together with NuRD at enhancers.

We do show that ZFP296 localises to promoters as well as enhancers, in agreement with the data mentioned by the reviewer. In general, in the revised manuscript we address the interaction between ZFP296 and SIN3A in more detail.

3. The ChIP-seq analysis in Figure 4 is not quantitative. ChIP-Rx for H3K27ac should be attempted in wt and KO cells and quantitative ChIP qPCR validations of Mbd3 and ChD4 should be performed at multiple different target sites in WT and KO ESCs.

We thank the reviewer for these excellent suggestions. We performed ChIP-seq with a spike-in for not only H3K27ac, but also MBD3, CHD4, and SIN3A in WT and *Zfp296* KO ESCs. In addition to showing that H3K27ac as well as SIN3A levels are not significantly altered between these two cell types, these experiments revealed that genome-wide NuRD binding actually decreases rather than increases.

4. The authors should validate the results in Figure 4 with two independent Zfp296- KO ESC clones.

We performed the spike-in ChIP-seq in two independent *Zfp296* KO ESC clones, as shown in Fig. 4 and Supplementary Fig. 3.

5. Western blots and/or RT-qPCR of Zfp296 KO cells should be included in Figure 4/Supplemental Figure 3 to confirm the knockout.

Unfortunately we were not able to obtain a working antibody for ZFP296. Instead, we now validated our knockouts with mass spectrometry.

Minor comments

6. The authors should clarify in the text and legend what "Cluster 1, 2 and 3" are in Figure 1.

In Figure 1, we have abolished the use of "Cluster 1, 2 and 3" in favour of a more informative description.

7. The authors should clarify why Mbd3 (the bait) is presenting as a high fold enrichment in only NPCs and not ESCs in Figure 2(d).

We think this observation is an artefact of the BAC lines we are using. As can be seen in Supplementary Fig. 2a, expression of endogenous MBD3 is lower in NPCs compared to ESCs, while the expression of MBD3-GFP increases a bit. This means that more GFP-MBD3 than endogenous MBD3 is incorporated into NuRD in NPCs, leading to a higher fold enrichment in the GFP pull-downs (cf. Fig 2a and b).

8. The yellow ChIP-seq plots are difficult to see on a print out of the paper.

We thank the reviewer for bringing this to our attention. We have now changed the yellow colour such that it is better visible on a print out of the paper.

Reviewer #3

I read with great interest the manuscript by Kloet et al. entitled "NuRD-interacting protein Zfp296 regulates genome-wide NuRD localization and differentiation of mouse embryonic stem cells". In this manuscript the authors use proteomic and genomic approaches to study NuRD complex dynamics during ESC differentiation into NPCs. They uncovered Zfp296 as novel NuRD interacting protein in ESC that seems to globally restrain NuRD binding in ESC and to contribute to proper ESC differentiation.

The role of the NuRD complex in ESC and its importance for pluripotency has been extensively studied. However, the role of NuRD during differentiation, the dynamics of its genomic targets and its cell-type specific interacting partners are much less studied. Therefore, in principle, this manuscript addresses an interesting question. However, in the current format I am afraid I can not support publication in Nature Communications due to some important concerns:

Major points

1. Although the authors acknowledge that NuRD composition, genomic targets and functional role has been extensively studied in ESC, they still decided to focus on how NuRD function is altered in ESC, in this case in the absence of Zfp296. Why not focusing on some of the NPC specific NuRD interacting proteins?. This could have made the study considerably more novel and original.

The reviewer is correct that such a study would have been very interesting; however, no novel NPC-specific NuRD interacting proteins were identified in our study (see Fig. 2).

2. The authors provide very limited mechanistic insights into how Zfp296 controls NuRD genomic targeting and/or enzymatic activity. They describe some potentially interesting observations, which unfortunately do not follow at all: (i) Zfp296 interacts with NuRD in the nucleoplasm rather than on chromatin, which could suggest that Zfp296 controls sub-cellular NuRD localization. Rather than following this idea, they hypothesize that Zfp296 binds to regulatory elements and recruits NuRD to such genomic targets. If this would be the case, then why Zfp296 preferentially interacts with NuRD in the nucleoplasm?. (ii) Based on ChIP-seq for Zfp296, the most overrepresented motif is the telomere repeat, which could be a potentially very interesting observation. However, this is barely mentioned in the rest of the manuscript. Are telomeres frequently bound by Zfp296 and NuRD?. Are they co-bound (sequential ChIPs)?. How is NuRD binding to telomeres affected in Zfp296 KO mESC?. Are telomeres affected in Zfp296 KO mESC and could this affect their pluripotency and differentiation potential?.

It is indeed true that we found that the stoichiometry of ZFP296 compared to MBD3 is higher in the nuclear extract than in the nuclear pellet fraction (Supplementary Fig. 2e). However, the nuclear extract does not represent the fraction that is not bound to chromatin at all, but rather the fraction that is more loosely bound to chromatin compared to the chromatin pellet fraction. This could indicate that ZFP296-NuRD binding to chromatin is more transient rather than stable, which points to dynamic regulation.

We indeed observed the telomere repeat region to be enriched under ZFP296 ChIP-seq peaks, but did not go into this further because we couldn't offer any more biological explanations. Also, our new experiments using drosophila spike-in chromatin and analyses do not support dynamic NuRD binding to repetitive elements upon a loss of ZFP296. We discuss this point further in the discussion.

3. The authors claim that the loss of Zfp296 in ESC globally affects H3K27ac. However, this claim is not totally supported by their data, which is not clearly presented. The differences shown in Fig4D seem rather minor and the shape of the H3K27ac signal is weird, as it does not show the typical dip in the middle. How many replicates did the author use?. Are the presented differences statistically significant?. The authors should present

H3K27ac data as heatmaps to appreciate whether all H3K27ac loci are moderately affected or, alternatively, whether a few loci are more severely affected?. Are the Zfp296 bound loci more sensitive to changes in H3K27ac and/or NuRD than other H3K27ac loci?. Last but not least, how do the authors interpret an increase in H3K27ac with a concomitant increase in NuRD binding in the Zfp296 KO ESC?. Are HDAC1/2 also being recruited to these loci in the Zfp296 KO ESC?.

Following the suggestion of another reviewer, we performed ChIP-seq using a spike-in for H3K27ac in WT and KO ESCs. Based on these analyses, we conclude that the previous change we saw in H3K27ac was not significant; indeed, our new data do not show such a difference. We now present these data more clearly by centring on the ZFP296 binding sites.

4. The characterization of the differentiation defects of Zfp296 KO ESC seems rather superficial, preliminary and correlative?. Are the genes showing differential expression in Zfp296 KO ESC preferentially bound by Zfp296?. Do these genes show preferential changes in H3K27ac and/or NuRD binding in Zfp296 KO ESC?. Their data upon embryoid body differentiation suggests a failure to differentiate: is this due to Zfp296 being directly involved in activation of lineage identity genes or, alternatively, due to a direct role of Zfp296 in the silencing of pluripotency genes?. In this regard, evaluating only Pou5f1 and Rex1 is clearly insufficient.

We thank the reviewer for this suggestion. We performed the suggested analysis, and found that of the genes whose transcript expression significantly changed, 17% were bound by MBD3, and 23% were bound by ZFP296 in wild-type cells.

The observed phenotype of delayed differentiation is in agreement with a study that was recently published as a pre-print on bioRxiv (<http://dx.doi.org/10.1101/269811>), which furthermore provides more insight into the underlying mechanism. We therefore chose not to elaborate on this aspect.

Minor comments

5. In Fig 1E, histone modification signals in each cluster should be shown in both ESC and NPC.

We thank the reviewer for this suggestion, and we have changed Fig. 1e accordingly.

6. The ontology terms shown in Supp Fig 1, they do not seem to be as different between the different NuRD binding sites as the authors claim in the text. Developmental terms dominate in all three categories, without an enrichment of pluripotency or neural terms in either the ESC or NPC-specific NuRD binding sites.

Since we adjusted Fig. 1c-e, we also repeated the GO term enrichment analysis with the new clusters. These results show more clearly that while the ESC- and NPC-enriched peaks are associated with developmental terms, the shared peaks are associated with more general housekeeping terms.

7. Fig 3A-C. The authors should perform sequential ChIP followed by qPCR analysis to investigate whether Zfp296 and NuRD subunits are simultaneously bound to their shared genomic targets.

We thank the reviewer for this suggestion and tried to perform the proposed experiment, but unfortunately were not able to get this to work technically. We think this is due to the low abundance of these factors as well as the low ChIP efficiency for transcription factors.

Reviewer #4

Kloet and colleagues describe in this paper the protein Zfp296 as a new interactor of the

NuRD complex in mouse embryonic stem cells. This finding is well supported by different experimental approaches and provide new insights into the function of the protein Zfp296. However, there are several aspects of the manuscript in need of some revision before the manuscript is suitable for publication.

Major comments

1. The authors should clarify why they assign all the effect on acetylation upon Zfp296 KO to the NuRD complex when Zfp296 is a shared interactor of the NuRD and Sin3/Hdac complexes. They should also perform ChIP on a component of the Sin3/Hdac complex in WT and upon Zfp296 to see the effect in this complex. As a general comment, even if the focus of the manuscript is on the effect of Zfp296 on the NuRD complex, the role of Sin3/HDAC in the story should be addressed in more detail because of the shared Zfp296 interaction.

We thank the reviewer for these suggestions. In our revised manuscript, we performed ChIP-seq for SIN3A and address the interplay between ZFP296 and the SIN3A complex in more detail.

2. Line 170: The authors should perform ChIP on one or two specific components of the Sin3/HDAC complex to see if distribution of this complex is also affected by the KO of Zfp296 in ESCs or NPCs.

We thank the reviewer for this suggestion. We have performed ChIP-seq on SIN3A in WT and Zfp296 KO ESCs, the results of which are described in the revised manuscript.

3. Lines 88-90: The sentence "In contrast, cell-type specific NuRD binding sites mostly map to intergenic and intronic regions which are marked with H3K27ac and H3K4me1, indicating that dynamic NuRD binding likely occurs at active enhancers (Fig. 1c-e)" is only applicable to NPC. For the ESC cell (Fig 1 Panel 2), the H3K4me3 is as higher as the H3K4me1.

We thank the reviewer for pointing out this ambiguity, and have changed the figures and the text accordingly.

4. Figure 1e: The authors should also comment on the differences in the distribution of the histone marks relative to the Mbd3 site in ESC and NPC (double peak depleted in the centre for cluster 2 versus single centred peak for cluster 3). Are these findings reproducible when analysing the Chd4 ChIP data the same way as represented in Fig 1 d/e? Is this dual distribution due to the Sox2/Oct4 binding motifs for ESC and Fos/Jun for NPC? Are there other factors described showing this binding pattern?

This is interesting indeed, as we mention in the text now, although we could not elucidate any biological mechanism behind this. However, recent data on H3K27ac at NuRD-bound loci also found that a single or double H3K27ac peak is characteristic of promoters or active enhancers, respectively (Bornelöv *et al.*, 2018, Mol. Cell), supporting our conclusions.

5. Lines 173-176: Is the observed difference in Fig4d and e significant? The mentioned increment is around 0.1 mean tags/bp. Is this enough to build the argument that follows this part?

Following the suggestion of another reviewer, we performed ChIP-seq using a spike-in for H3K27ac in WT and KO ESCs. Based on these analyses, we conclude that the previous change we saw in H3K27ac was not significant; indeed, our new data do not show such a difference. We have changed the text accordingly.

Minor comments

6. Line 61-62: Explain the developmental connection between ESC and NPC, or refer it to an existing paper/review.

Adjusted.

7. Line 70: Define LIF

Corrected.

8. Line 84: Give a number/percentage instead of "many"

Adjusted.

9. Line 88: Give a number/percentage instead of "mostly"

Adjusted.

10. Line 122: Rephrase "Zfp296 is relatively uncharacterized protein". Describe what is known and what is unknown or unusual to transmit a more defined idea of the knowledge on Zfp296.

We already provide some information about ZFP296 at this point, and go into it further in the discussion.

11. Line 136: Which is the stoichiometry of the NuRD and the Sin3/Hdac complexes normalized to the bait in the Zfp296 IP? Could the authors estimate from there which is the fraction of Zfp296 interacting with each complex? This may help to answer the Major Comment 1?

These stoichiometry data are depicted in Supplementary Fig. 2h, which is now normalised to the GFP-ZFP296 bait. However, from such data it is not possible to accurately assess what the fraction of ZFP296 interacting with each complex is, also because these complexes share some subunits.

12. Line 137/138: "the core NuRD complex is quite stable". Define the stability of the complex using data. In line 45 the proteins Cdk2ap1/Doc1 and Mbd3 are described as part of the NuRD core complex. However, in Fig2d, it seems that Cdk2ap1/Doc1 and Mbd3 are enriched in the NPC when compared to the ESC, as this will point to different composition of the complex depending on the cell state. The authors should comment on that. Plotting the comparison between Mbd3-GFP in mESC and NPC to see the changes of all the NuRD core complex components will help to visualize possible changes in complex composition.

We thank the reviewer for bringing this to our attention. There are indeed some changes in the composition of the NuRD complex, which could be an artefact of the BAC system, but we have no reason to assume this greatly changes its biology. We have now changed the text to avoid any confusion. In addition, we replaced Fig. 2d with a new figure (now Supplementary Fig. 2b), in which we plot the LFQ values from the MBD3-GFP interactors obtained in the GFP pulldowns from Fig. 2a and b, as suggested by the reviewer.

13. Line 143 "many of these overlap" give a number.

Adjusted.

14. Line 144: "the most enriched DNA sequence motif under Zfp296 peaks is TTAGGG, which is the telomere repeat motif (Fig. 3d)." The authors should mention if is there any

known telomere interacting protein enriched in the Zfp296 IP and highlight it in the corresponding plot (Figure 2e).

We did not identify such a protein in the GFP-ZFP296 pulldowns.

15. Line 149: "Zfp296 binding sites map to a mix of transcription start sites (26%), intergenic (39%) and intronic regions (30%)" The authors should compare these percentages with the ones of TSS, intergenic and intronic regions in the complete genomic, to see if the distribution is normal or enriched for any of the three classes.

We thank the reviewer for this suggestion, and have added this analysis.

16. Line 150/153/161: Give a number for "large majority", "small percentage" and "a global increase".

Rewritten.

17. Line 151: Fig3g should also have the same kind of plotting that Fig 1e is representing for the ChIP Mbd3 data.

We thank the reviewer for this suggestion, and we have changed Fig. 3g accordingly (Fig 3e in the revised manuscript).

18. Linen 190: The authors should give both the absolute and the percentage of differentially expressed genes/proteins in the Zfp296 KO vs WT ESC comparison.

Adjusted.

19. Line206: Comments on the role of Sin3/Hdac.

We now address the interplay between ZFP296 and SIN3A in more detail.

20. Line 372: "Statistical cut-offs were chosen such that no proteins were present as outliers on the control, non-GFP side of the volcano plot." Volcano plot Fig2b is not following the rule described here.

We thank the reviewer for bringing this to our attention, and we have changed Figure 2b accordingly.

21. Line 407: Submission of the data should be done and PDX number should be added before accepting the manuscript.

Upload of the data to PRIDE and GEO in ongoing and will be completed before final acceptance of the paper

22. Fig1: The number of peaks in the intersection of the Venn diagrams should be the same as the cluster 1 shared from Panel c. Otherwise explain the relationship between the numbers from Panel a and c. Scale of plots from Panel e should be equal to allow better comparison. Why are the numbers in cluster 1 from Panel e so high when only 541 peaks are used, and for clusters 2 and 3 so low (0.75 and 0.2 max) when the number of peaks are 1149 and 1607?

We thank the reviewer for bringing this to our attention. This discrepancy was caused by us considering only one MBD3 ChIP-seq replicate for the one figure while considering two for the other. We now changed all figures to agree with Fig. 1a which is based on 2 replicates.

We also changed the y-axis of Fig. 1e to be equal across all plots.

23. Fig2: The subunits of the Nurd complex in Panels a and b and the subunits of Nurd and Sin3 complexes in Panel e should be colour coded. The original data from the bar chart in panel d should be also displayed as volcano plots to get a better overview of the differences between ESC and NPC changes. In this volcano plot the subunits of Nurd and Sin3 should be colour coded as well.

We thank the reviewer for these suggestions. We now colour coded Fig. 2a-d, as well as Supplementary Fig. 2b and f. In addition, we introduced two new figures (Fig. 2c and Supplementary Fig. 2b), which we hope give a better overview of the differences between ESC and NPC.

Reviewer #1 (Remarks to the Author):

The manuscript has improved by the inclusion of additional data (p300 ChIPseq data to strengthen the conclusion that ESC-specific NuRD binding sites are enriched at enhancers; SIN3A ChIPseq to evaluate the effect of ZFP296 KO on SIN3A complex recruitment) and by displaying data in different formats (e.g. centering ChIPseq reads also on ZFP296 binding sites).

Most importantly, the manuscript has improved by including a spike in control in the ChIPseq assays. Remarkably, this has significantly changed two of the main conclusions: First, the authors no longer claim that NuRD binding to chromatin increases in response to ZFP296 KO. Rather, they report a genomewide decrease of MBD3 and CHD4 binding as well as a decrease over ZFP296 binding sites. Moreover, using the same protocol the previously reported increase in H3K27ac is no longer observed. However, I do note that the latter is true only for one of the two ZFP296 KO ESC lines analysed. An increase in K27ac is still apparent in the ESC KO line 27 (Sup Figure 3f). I think the authors should discuss on this variability. In a similar vein, the effect of ZFP296 KO on SIN3A chromatin association is somewhat different between the two ESC KO lines analysed. In one case (main figure) there actually is a decrease in binding that is comparable to the decrease observed for MBD3. The decrease in MBD3 binding interpreted as demonstrating reduced NuRD association but the similar decrease in SIN3A ChIPseq signal is interpreted as not being a significant change. In the second ESC KO clone (sup figure) there is indeed no discernable effect on K27ac levels. Again, the authors should to comment on this variability and rephrase their conclusions accordingly.

In summary, my concerns have been fully addressed in the revised version of the manuscript or, in some cases, made irrelevant by the authors now drawing opposite conclusions based on their new data. The findings now support a more conventional model, where ZFP296 contributes to the recruitment of NuRD to chromatin. Nevertheless, the manuscript still makes an important contribution to the field and establishes ZFP296 as an important regulator of chromatin and the NuRD complex that is required for efficient differentiation of ESCs.

Reviewer #2 (Remarks to the Author):

In the revised paper, entitled “NuRD-interacting protein Zfp296 regulates genome wide NuRD localisation and differentiation of mouse embryonic stem cells”, the authors addressed most of the points raised and thus improved the paper.

Although the authors have not found a working antibody for Zfp296, validation of Zfp296 KO by mass spec (Fig 4a and FigS3a&b) is sufficient to confirm knockout. This was not apparent and it may be helpful to highlight Zfp296 in this more clearly.

The most dramatic change in the paper came from the results from the spiked-in ChIP-seq. This has changed the narrative of the story and revealed a small decrease in NuRD binding, rather than an increase as initially reported, in the Zfp296 KO cells. This new finding highlights the importance of quantitative ChIP-Rx. The authors should now elaborate the moderate reduction in NuRD binding by including some examples of these decreases on some example gene loci, coupled with Zfp296 binding at these same sites.

Importantly, they should further analyze their datasets bioinformatically to carefully evaluate if NuRD binding decreases more dramatically at Zfp296 sites compared to non-Zfp296 sites. This would strengthen and clarify the paper's new narrative and perhaps support the model that Zfp296 contributes to targeting NuRD at some of its binding sites, but not all.

Reviewer #3 (Remarks to the Author):

After reading the revised manuscript version by Kloet et al. I am afraid that most of my suggestions and concerns have not been properly addressed. The main changes in the manuscript are the new quantitative ChIP-seq experiments for NuRD and H3K27ac. The authors now observe decreased NuRD binding in the Zfp296 KO cells but no changes in H3K27ac at neither NuRD nor Zfp296 sites. However, the reported changes in NuRD binding, especially for MBD3 are barely visible (see heatmaps in Fig 4b), yet they are not quantified in any way to support their significance. Moreover, the lack of effects on H3K27ac levels and on the expression of most genes suggest that, if real, the decrease NuRD binding in the Zfp296 KO ESC is not of much biological relevance. Furthermore, previous reports using MBD3 KO ESC (Reynolds et al, 2012) observed increased expression of naïve pluripotency genes, which are not seen at all in Zfp296 KO ESC. Lastly, upon LIF removal and exit from pluripotency, the ZFP296 KO cells displayed decreased induction of lineage specifiers but proper silencing of pluripotency genes. This is again different from previous observations for MBD3 KO ESC, which upon LIF removal failed to properly silence pluripotency genes (Reynolds et al., 2013). Overall, the provided data does not provide strong support to claim that Zfp296 function is mediated through the recruitment of NuRD to regulatory elements. Zfp296 seems to have an important function during exit of pluripotency and germline specification (Hackett et al, 2018), but in my opinion the manuscript fails to convincingly show the mechanistic basis of such function, which can certainly include NuRD-independent mechanisms.

Reviewer #4 (Remarks to the Author):

The authors sufficiently addressed all my concerns. I consider the manuscript now ready for publication in Nat comm

We would like to thank the reviewers for their careful reading of our revised manuscript, and are pleased that most reviewers are content with the improvements we made. In addition, we now further adjusted our manuscript based on the new suggestions that they offered. Below, we provide a point-to-point response to the reviewer comments, where the original comments are in italics and our response is in regular font.

Reviewer #1:

The manuscript has improved by the inclusion of additional data (p300 ChIPseq data to strengthen the conclusion that ESC-specific NuRD binding sites are enriched at enhancers; SIN3A ChIPseq to evaluate the effect of ZFP296 KO on SIN3A complex recruitment) and by displaying data in different formats (e.g. centering ChIPseq reads also on ZFP296 binding sites). Most importantly, the manuscript has improved by including a spike in control in the ChIPseq assays.

1. Remarkably, this has significantly changed two of the main conclusions: First, the authors no longer claim that NuRD binding to chromatin increases in response to ZFP296 KO. Rather, they report a genomewide decrease of MBD3 and CHD4 binding as well as a decrease over ZFP296 binding sites. Moreover, using the same protocol the previously reported increase in H3K27ac is no longer observed. However, I do note that the latter is true only for one of the two ZFP296 KO ESC lines analysed. An increase in K27ac is still apparent in the ESC KO line 27 (Sup Figure 3f). I think the authors should discuss on this variability. In a similar vein, the effect of ZFP296 KO on SIN3A chromatin association is somewhat different between the two ESC KO lines analysed. In one case (main figure) there actually is a decrease in binding that is comparable to the decrease observed for MBD3. The decrease in MBD3 binding interpreted as demonstrating reduced NuRD association but the similar decrease in SIN3A ChIPseq signal is interpreted as not being a significant change. In the second ESC KO clone (sup figure) there is indeed no discernable effect on K27ac levels. Again, the authors should comment on this variability and rephrase their conclusions accordingly.

We thank the reviewer for bringing this to our attention. As can be seen in Fig. 4a and Supplementary Fig. 3a, there is some variability in the depletion of ZFP296 between the different knockout lines. While this could explain the inconsistency in the data for H3K27ac and SIN3A, the findings for MBD3 and CHD4 are reproducible between the two cell lines. We therefore think that the changes in NuRD binding are genuine biological effects, but we are more careful with drawing conclusions from the H3K27ac and SIN3A data. We adjusted the text to better reflect our considerations, both technical and biological. In addition, we now present our data as boxplots (Fig. 4f and Supplementary Figs. 3f and h), which makes it easier to appreciate the differences between the different cell lines.

In summary, my concerns have been fully addressed in the revised version of the manuscript or, in some cases, made irrelevant by the authors now drawing opposite conclusions based on their new data. The findings now support a more conventional model, where ZFP296 contributes to the recruitment of NuRD to chromatin. Nevertheless, the manuscript still makes an important contribution to the field and establishes ZFP296 as an important regulator of chromatin and the NuRD complex that is required for efficient differentiation of ESCs.

We thank the reviewer for her/his support and the current as well as previous suggestions, which helped us to improve the manuscript.

Reviewer #2:

In the revised paper, entitled "NuRD-interacting protein Zfp296 regulates genome wide NuRD localisation and differentiation of mouse embryonic stem cells", the authors addressed most of the points raised and thus improved the paper.

1. Although the authors have not found a working antibody for Zfp296, validation of Zfp296 KO by mass spec (Fig 4a and FigS3a&b) is sufficient to confirm knockout. This was not apparent and it may be helpful to highlight Zfp296 in this more clearly.

We thank the reviewer for pointing this out to us, and we now applied colour-coding to Fig. 4a and Supplementary Figs. 3a and b to clarify this.

2. The most dramatic change in the paper came from the results from the spiked-in ChIP-seq. This has changed the narrative of the story and revealed a small decrease in NuRD binding, rather than an increase as initially reported, in the Zfp296 KO cells. This new finding highlights the importance of quantitative ChIP-Rx. The authors should now elaborate the moderate reduction in NuRD binding by including some examples of these decreases on some example gene loci, coupled with Zfp296 binding at these same sites.

The reviewer is correct that ChIP-Rx turned out to be essential for our story, and we would like to thank her/him again for this excellent suggestion. As suggested, we now included some screenshots of example ZFP296-bound genomic loci where this moderate decrease in NuRD binding can be appreciated (Supplementary Fig. 3g).

3. Importantly, they should further analyze their datasets bioinformatically to carefully evaluate if NuRD binding decreases more dramatically at Zfp296 sites compared to non-Zfp296 sites. This would strengthen and clarify the paper's new narrative and perhaps support the model that Zfp296 contributes to targeting NuRD at some of its binding sites, but not all.

We thank the reviewer for this suggestion. We now included the suggested analysis (Fig. 4e and Supplementary Fig. 3e), which indeed indicates that co-localisation with ZFP296 is a predictor for the difference in MBD3 signal between WT and Zfp296 KO cells.

Reviewer #3:

After reading the revised manuscript version by Kloet et al. I am afraid that most of my suggestions and concerns have not been properly addressed.

1. The main changes in the manuscript are the new quantitative ChIP-seq experiments for NuRD and H3K27ac. The authors now observe decreased NuRD binding in the Zfp296 KO cells but no changes in H3K27ac at neither NuRD nor Zfp296 sites. However, the reported changes in NuRD binding, especially for MBD3 are barely visible (see heatmaps in Fig 4b), yet they are not quantified in any way to support their significance.

We are aware that quantitative differences are difficult to see when data are displayed as a heatmap (Fig. 4b), which is why we displayed the same data also as band plots (Fig. 4c), where it is easier to appreciate such differences. In addition, we now also present our data as boxplots to better quantify these differences (Figs. 4d,f and Supplementary Figs. 3d,f,h).

2. Moreover, the lack of effects on H3K27ac levels and on the expression of most genes suggest that, if real, the decrease NuRD binding in the Zfp296 KO ESC is not of much biological relevance.

We indeed observe modest effects on the transcriptome and proteome upon knockout of Zfp296, which is in agreement with modest effects on NuRD binding. We therefore conclude that ZFP296 contributes to, but is not solely responsible for, recruitment of NuRD to ESC-specific binding sites. Nevertheless, we show that Zfp296 knockout has an effect on the differentiation potential of ESCs, indicating its biological relevance.

3. Furthermore, previous reports using MBD3 KO ESC (Reynolds et al, 2012) observed increased expression of naïve pluripotency genes, which are not seen at all in Zfp296 KO ESC. Lastly, upon LIF removal and exit from pluripotency, the ZFP296 KO cells displayed decreased induction of lineage specifiers but proper silencing of pluripotency genes. This is again different from previous observations for MBD3 KO ESC, which upon LIF removal failed to properly silence pluripotency genes (Reynolds et al., 2013).

We indeed did not observe increased expression of naïve pluripotency genes or disturbed silencing of pluripotency genes upon differentiation in Zfp296 KO ESCs. However, the cited work (Reynolds et al., 2012) proposes a model in which Mbd3 KO ESCs fail to differentiate, which certainly agrees with our Zfp296 KO ESCs showing a delay in differentiation. In addition, it is important to keep in mind that while MBD3 is a core component of the NuRD complex, we show that ZFP296 is a substoichiometric interactor that contributes to NuRD binding. It is thus only expected that Zfp296 KO ESCs show a milder phenotype than Mbd3 KO ESCs. Lastly, we would like to emphasise that conflicting evidence exists for the role of MBD3/NuRD in pluripotency, as we mention in our discussion. It is thus clear that more research is needed to elucidate this issue, to which we think that our paper contributes.

4. Overall, the provided data does not provide strong support to claim that Zfp296 function is mediated through the recruitment of NuRD to regulatory elements. Zfp296 seems to have an important function during exit of pluripotency and germline specification (Hackett et al, 2018), but in my opinion the manuscript fails to convincingly show the mechanistic basis of such function, which can certainly include NuRD-independent mechanisms.

We agree that our revised data no longer support the claim that ZFP296 recruits NuRD to regulatory elements, which is why we removed this suggestion from the manuscript. However, we do think there is sufficient evidence that ZFP296 contributes to recruitment of NuRD to ESC-specific sites, and that this is important for the differentiation potential of the cells. We adjusted our manuscript to further emphasise this.

Reviewer #4:

The authors sufficiently addressed all my concerns. I consider the manuscript now ready for publication in Nat comm.

We thank the reviewer for her/his support and previous suggestions and comments, which helped us to improve the manuscript.

Reviewer #1 (Remarks to the Author):

My concerns had been fully addressed after the first revision. Also, the suggestions I made about the revised version have been considered. I am perfectly happy with the manuscript being published in its present form.

Reviewer #2 (Remarks to the Author):

The authors sufficiently addressed all my suggestions and thereby improved the paper. Reviewer 3 is pointing out that the loss of Zfp296 does not strongly reduce NuRD binding and this is a fair point. However, I believe the authors adequately modified their claims and text accordingly.

In light of the dramatic changes with the NURD quantitative ChIP-RX, it raises a similar question about their data and claims related to SIN3A and K27ac. It could well be that loss of Zfp296 does affect Sin3a complex binding, because the data in Figure S3h is not based on quantitative ChIP-seq (ChIP-RX) of Sin3a. Are their data on H2K27ac done by quantitative ChIP-RX? If not, they should modify their discussion on these data. If the experiment is not with spike-in, then they simply can not say anything.

Reviewer #3 (Remarks to the Author):

The authors have now sufficiently addressed my concerns and, thus, I recommend the manuscript for publication.

We would like to thank the reviewers for their careful reading of the latest version of our revised manuscript, and their constructive suggestions and comments that helped us to get to this version. Below, we provide a point-to-point response to the reviewer comments, where the original comments are in italics and our response is in regular font.

Reviewer #1:

My concerns had been fully addressed after the first revision. Also, the suggestions I made about the revised version have been considered. I am perfectly happy with the manuscript being published in its present form.

We thank the reviewer for her/his support.

Reviewer #2:

The authors sufficiently addressed all my suggestions and thereby improved the paper. Reviewer 3 is pointing out that the loss of Zfp296 does not strongly reduce NuRD binding and this is a fair point. However, I believe the authors adequately modified their claims and text accordingly.

We thank the reviewer for her/his support.

In light of the dramatic changes with the NURD quantitative ChIP-RX, it raises a similar question about their data and claims related to SIN3A and K27ac. It could well be that loss of Zfp296 does affect Sin3a complex binding, because the data in Figure S3h is not based on quantitative ChIP-seq (ChIP-RX) of Sin3a. Are their data on H2K27ac done by quantitative ChIP-RX? If not, they should modify their discussion on these data. If the experiment is not with spike-in, then they simply can not say anything.

As described on page 6, lines 200-204 and 207-211, ChIP for SIN3A and H3K27ac was also performed with a spike-in control.

Reviewer #3:

The authors have now sufficiently addressed my concerns and, thus, I recommend the manuscript for publication.

We thank the reviewer for her/his support.